# chTOG is a conserved mitotic error correction factor

**Jacob A Herman, Matthew P Miller[†], Sue Biggins***

Howard Hughes Medical Institute, Division of Basic Sciences, Fred Hutchinson Cancer Research Center, Seattle, United States

**Abstract** Accurate chromosome segregation requires kinetochores on duplicated chromatids to biorient by attaching to dynamic microtubules from opposite spindle poles, which exerts forces to bring kinetochores under tension. However, kinetochores initially bind to microtubules indiscriminately, resulting in errors that must be corrected. While the Aurora B protein kinase destabilizes low-tension attachments by phosphorylating kinetochores, low-tension attachments are intrinsically less stable than those under higher tension in vitro independent of Aurora activity. Intrinsic tension-sensitive behavior requires the microtubule regulator Stu2 (budding yeast Dis1/XMAP215 ortholog), which we demonstrate here is likely a conserved function for the TOG protein family. The human TOG protein, chTOG, localizes to kinetochores independent of microtubules by interacting with Hec1. We identify a chTOG mutant that regulates microtubule dynamics but accumulates erroneous kinetochore-microtubule attachments that are not destabilized by Aurora B. Thus, TOG proteins confer a unique, intrinsic error correction activity to kinetochores that ensures accurate chromosome segregation.

**\*For correspondence:**
sbiggins@fredhutch.org

**Present address:** [†]Department of Biochemistry, University of Utah, Salt Lake City, United States

**Competing interests:** The authors declare that no competing interests exist.

## Introduction

Eukaryotic cell division requires the duplication and accurate segregation of up to hundreds of chromosomes. In most species, chromosome segregation is carried out by a conserved network of dozens of kinetochore factors that assemble into a megadalton protein complex on centromeres to link chromosomes and dynamic microtubule polymers (*Hara and Fukagawa, 2018*). During mitosis, the microtubule cytoskeleton is organized into a bipolar spindle such that each and every pair of duplicated sister chromosomes becomes bioriented (attached to microtubules anchored to opposite spindle poles). In this state, coordinated depolymerization of all kinetochore-bound microtubules results in the accurate segregation of every chromosome. However, the biorientation process is error-prone, as early in mitosis both the mitotic spindle and the chromosomes lack organization, yet kinetochores begin forming attachments (*Cimini et al., 2003*; *Kapoor et al., 2000*; *Maiato et al., 2017*). While some kinetochore pairs become properly bioriented, others attach to microtubules emanating from the same spindle pole and must be destabilized. It is well appreciated that tension generated by depolymerizing microtubules pulling across a pair of bioriented kinetochores is a key signal that attachments should be stabilized (*Akiyoshi et al., 2010*; *King and Nicklas, 2000*; *Salmon and Bloom, 2017*). This biochemical error correction system is primarily known to be regulated via the Aurora B protein kinase and its downstream targets, which specifically *destabilize* low-tension kinetochore-microtubule attachments (*Biggins and Murray, 2001*; *Cheeseman et al., 2006*; *DeLuca et al., 2006*; *Liu et al., 2009*). However, we recently demonstrated that the protein Stu2 (yeast member of the Dis1/XMAP215 family) confers tension-sensitive binding behaviors to reconstituted yeast kinetochore-microtubule attachments (*Akiyoshi et al., 2010*; *Miller et al., 2016*). Moreover, this intrinsic tension-dependent activity functioned completely independent of Aurora B activity (*Akiyoshi et al., 2010*; *London et al., 2012*; *Miller et al., 2016*), suggesting that cells have multiple mechanisms to destabilize incorrect attachments.

Stu2 and the entire Dis1/XMAP215 family are well-characterized microtubule regulators that contribute to the nucleation, polymerization, and organization of the cytoskeleton and spindle in both developing and somatic cells (*Brouhard et al., 2008*; *Cullen et al., 1999*; *Gard and Kirschner, 1987*; *Kosco et al., 2001*; *Milunovic-Jevtic et al., 2018*; *Roostalu et al., 2015*; *Shirasu-Hiza et al., 2003*). This protein family is thought to accomplish these diverse forms of microtubule regulation through two regulatory regions. First, these proteins contain an array of 2–5 TOG domains that are each capable of binding an α/β tubulin dimer. Second, an unstructured 'basic patch' enriched for Lysine and Arginine residues appears to contribute to a non-specific electrostatic interaction with the negatively charged microtubule lattice (*Geyer et al., 2018*; *Wang and Huffaker, 1997*; *Widlund et al., 2011*). In vitro, these two regulatory regions catalyze the nucleation and elongation of microtubule polymers. However, Stu2's ability to confer tension-dependent binding behavior to reconstituted yeast kinetochore-microtubule attachments appears to be independent of its role in regulating microtubules, as all measures of dynamicity remained unchanged when Stu2 was absent from reconstitutions (*Miller et al., 2016*; *Miller et al., 2019*). Thus far, reconstitution experiments have been the only means to specifically study Stu2/XMAP215 regulation of kinetochore-microtubule attachments as in vivo depletion studies result in dominant defects in mitotic spindle organization and function (*Kosco et al., 2001*; *Miller et al., 2019*). We recently described a Stu2 mutant that supported spindle formation in yeast cells, but not biorientation, which provided in vivo evidence that Stu2 functions as an error correction factor independent of its role organizing the mitotic spindle (*Miller et al., 2019*). However, this mutant does not function as a microtubule polymerase in vitro (*Geyer et al., 2018*), raising the possibility that these two activities are connected in cells.

Similarly, depletion of the human ortholog, chTOG (TOG/TOGp/CKAP5), results predominantly in multipolar spindle assembly defects (*Cassimeris and Morabito, 2004*; *Gergely et al., 2003*). Chromosome biorientation is possible with partial depletion, and these kinetochores exhibit dampened oscillations and decreased inter-kinetochore tension (*Barr and Gergely, 2008*; *Cassimeris et al., 2009*). Although these data suggest chTOG regulates kinetochore-microtubule attachments, it is not clear if this role is related to regulating microtubule dynamics. Separation of these activities in human cells has also been limited by the ability to express mutant chTOG proteins. These large proteins (225 kDa) are inefficient to transduce through chemical and viral means, and negatively affect proliferation when overexpressed. Therefore, it has been assumed that TOG proteins regulate kinetochore-microtubule attachments via secondary effects on microtubule polymerization rates and it has been unclear if this protein family has a direct function in mitotic error correction in multicellular eukaryotes.

Here, we demonstrate that the Stu2-dependent error correction process observed in budding yeast is a conserved process in human cells. Similar to the yeast proteins (*Miller et al., 2016*), we found that chTOG associates with and requires the conserved microtubule binding factor Hec1 for kinetochore localization. Additionally, we show that a pair of point mutations in chTOG's 'basic linker' domain inhibits error correction activity but does not compromise its ability to regulate the microtubule cytoskeleton. Together, this work reveals that chTOG functions in an evolutionarily conserved manner to destabilize erroneous, low-tension attachments. Moreover, we find that Aurora B phosphoregulation of its key kinetochore substrate, Hec1/Ndc80, cannot compensate for loss of chTOG-mediated error correction. Our work further elucidates a largely uncharacterized, intrinsic mechanism by which kinetochores sense and respond to biomolecular forces in order to prevent errors in chromosome segregation.

## Results

### A pool of chTOG resides at kinetochores, independent of microtubule plus ends

Previous microscopy studies suggested that chTOG localizes to kinetochores (*Campbell et al., 2019*; *Gutiérrez-Caballero et al., 2015*; *Ryan et al., 2020*), similar to the budding yeast ortholog (*Miller et al., 2016*; *Miller et al., 2019*); however, it was unclear if this population was simply bound to microtubule tips. To address this, we used engineered HCT116 cells where the endogenous chTOG genes were epitope tagged with EGFP (*Cherry et al., 2019*) to determine whether chTOG specifically localizes to kinetochores throughout mitosis (*Figure 1a*). We found that chTOG is largely

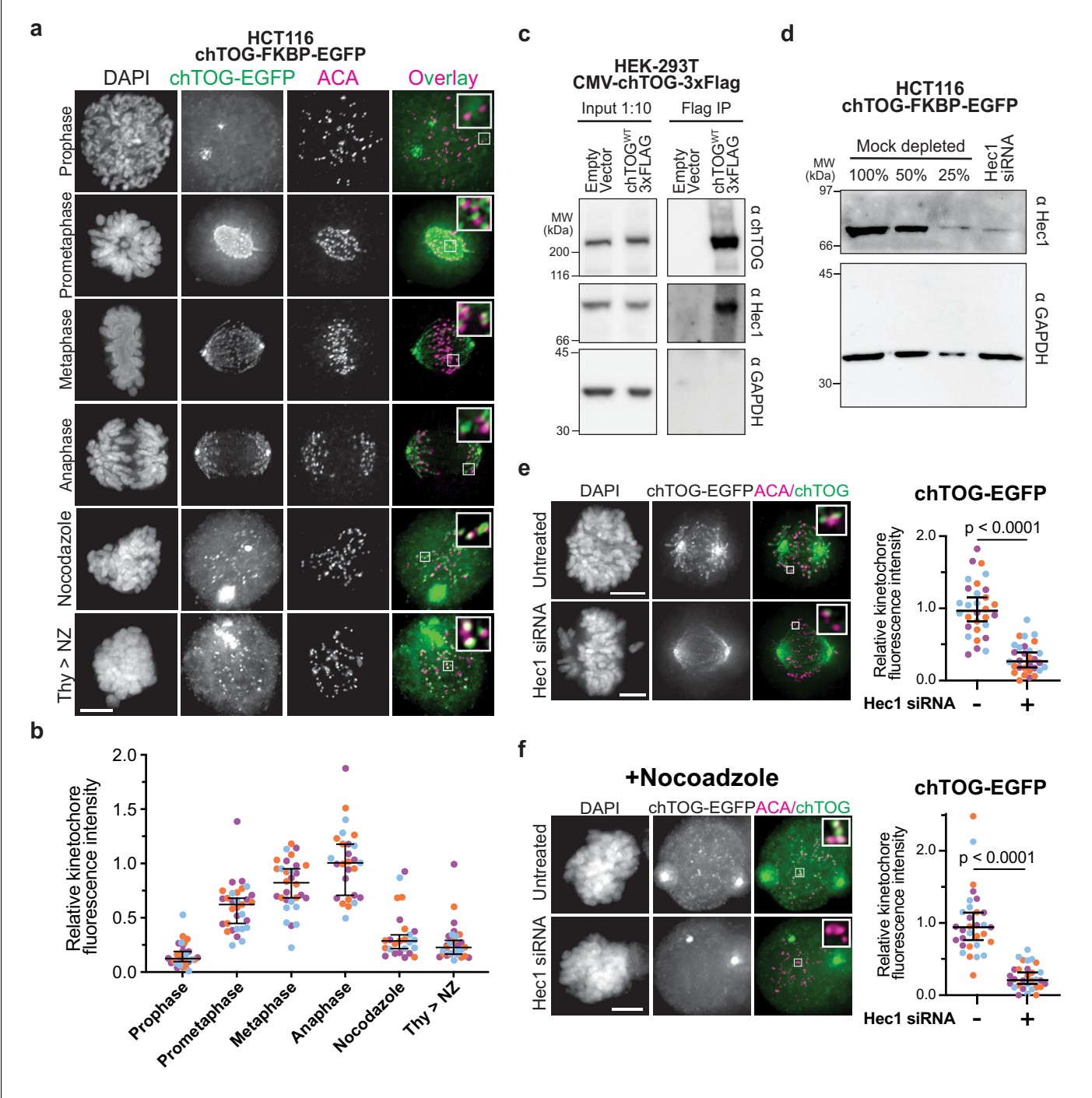

**Figure 1.** chTOG localizes to kinetochores during mitosis. (**a**) Immunofluorescence images of chTOG subcellular localization during mitosis, as visualized in HCT116 cells expressing endogenously epitope-tagged chTOG-EGFP. Anti-centromere protein antibody (ACA) staining marks the centromere-binding proteins and representative images are shown with inlays of kinetochore proximal chTOG at each stage of mitosis. (**b**) Quantification of chTOG kinetochore association. Each data point represents mean chTOG-EGFP fluorescence intensity at all kinetochores in a single cell normalized to the mean value of the anaphase population (**c**) HEK-293T cells with either an empty vector control or overexpressed chTOG-3Flag were immunoprecipitated using anti-Flag antibodies. Immunoblots of the input (left) or Flag IP (right) show that the endogenous Hec1/Ndc80 protein specifically co-purified with chTOG. Endogenous and epitope tagged chTOG cannot be individually resolved by anti-chTOG immunoblotting because the 3Flag tag increases the protein's predicted MW by only 3%. Anti-GAPDH served as a non-specific control. (**d**) Immunoblotting with anti-Hec1 antibodies was performed on samples of mock-depleted lysate that were diluted to contain the indicated percent of total protein and compared to a lysate prepared from a population of HCT116 cells treated with Hec1 siRNA. Greater than 75% of Hec1 protein was depleted in the siRNA-treated cells.

*Figure 1 continued on next page*

Figure 1 continued

Anti-GAPDH is a loading control. (e) Kinetochore localization of chTOG-EGFP in Hec1 depleted HCT116 cells was determined by fluorescence microscopy. Representative images are shown and were quantified on the right to show endogenously tagged chTOG-EGFP signal at kinetochores decreased by ~70% in siRNA-treated HCT116 cells. Each data point represents mean fluorescence intensity at all kinetochores in a single cell normalized to the mean value of the mock depleted population. (f) Same as (e) but cells were treated with 10 μM nocodazole for 1 hr prior to fixation to preclude any chTOG bound to the microtubule tips from this analysis. All scale bars are 5 μm; contrast on inlays was adjusted independently; data points on graphs are grouped from three experimental replicates and colored according to each replicate with median and 95% confidence intervals displayed in black. p-Values determined by an unpaired Mann-Whitney test.

The online version of this article includes the following figure supplement(s) for figure 1:

**Figure supplement 1.** Exogenously expressed chTOG-EGFP localizes to kinetochores in HeLa cells.
**Figure supplement 2.** Hec1 siRNA does not affect chTOG protein levels.

excluded from the nucleus until prometaphase, when it appeared on kinetochores as assayed by co-localization with anti-centromere antibody (ACA) (*Figure 1a*). As cells progress through mitosis, its kinetochore localization increased 2.5-fold (*Figure 1a,b*). To determine what fraction of this signal is specifically interacting with kinetochores and not microtubule tips, we treated cells with nocodazole to depolymerize microtubules. At least 60% of chTOG recruited to prometaphase kinetochores and 30% recruited to anaphase kinetochores is independent of microtubules (*Figure 1a,b*). This trend was also observed in HeLa cells overexpressing exogenous chTOG-EGFP (*Figure 1—figure supplement 1a*). To determine whether microtubule attachment delivers chTOG to kinetochores, we arrested cells in S phase with thymidine and released them into the cell cycle in the presence of nocodazole so that kinetochore-microtubule attachments never occurred. In this experiment, chTOG was still detected on kinetochores, consistent with a kinetochore-bound pool that is separate from microtubule tips (*Figure 1a,b*, *Figure 1—figure supplement 1a*). chTOG and its budding yeast ortholog, Stu2, physically interact with the Ndc80 kinetochore complex in vitro (*Miller et al., 2016*; *Zahm et al., 2020*). To test whether they associate in human cells, we immunopurified FLAG-tagged chTOG from HEK-293T cells under conditions refractory to microtubule formation and found that the endogenous Hec1 protein co-purifies (*Figure 1c*). We therefore tested whether the kinetochore-bound pool of chTOG depends on Hec1 by depleting Hec1 from cells and quantifying the chTOG-EGFP intensity proximal to kinetochores (*Figure 1d–f*, *Figure 1—figure supplement 1b–e*). We observed a 70% reduction in chTOG-EGFP signal (*Figure 1e*). However, because Hec1 depletion ablates kinetochore-MT attachments (*DeLuca et al., 2006*), this result was again confounded by chTOG's localization to both kinetochores and microtubule tips. Thus, we performed the same experiment in the presence of nocodazole to separate the two chTOG populations, and again observed a 70% reduction in chTOG-EGFP signal at kinetochores (*Figure 1f*). We confirmed that total protein levels of chTOG were unaffected by Hec1 siRNA (*Figure 1—figure supplement 2a*). These data are consistent with the Ndc80 complex serving as the primary receptor for chTOG. Therefore, a pool of chTOG localizes to kinetochores in a Hec1-dependent manner that is distinct from the chTOG at the microtubule plus-ends, suggesting a function for chTOG on kinetochores.

## Two residues in the basic linker are essential for viability in yeast and human cells

To determine the role of chTOG at the kinetochore, we required a mutant that specifically inhibited its kinetochore function without affecting the protein's numerous other microtubule-based activities. However, chTOG is an extremely large, multidomain protein consisting of 2032 residues, making it difficult to identify a separation of function mutant. It regulates microtubule dynamics using an array of five TOG domains (*Charrasse et al., 1998*; *Charrasse et al., 1995*; *Spittle et al., 2000*; *Figure 2a*). Additionally, chTOG contains a flexible 'basic linker' region that in vitro experiments suggest provides a non-specific, electrostatic interaction with the negatively charged microtubule lattice to facilitate diffusion to the plus-end (*Geyer et al., 2018*; *Wang and Huffaker, 1997*; *Widlund et al., 2011*). Finally, there is a small domain within the C-terminus of chTOG that serves as a protein interaction hub to mediate its various intracellular localization patterns (*Gutiérrez-Caballero et al., 2015*; *Hood et al., 2013*; *van der Vaart et al., 2011*). All these protein elements are present in the budding yeast ortholog Stu2, which contains just two TOG domains but homodimerizes through a coiled-coil (CC) region (*Figure 2a*). We therefore took advantage of yeast genetic

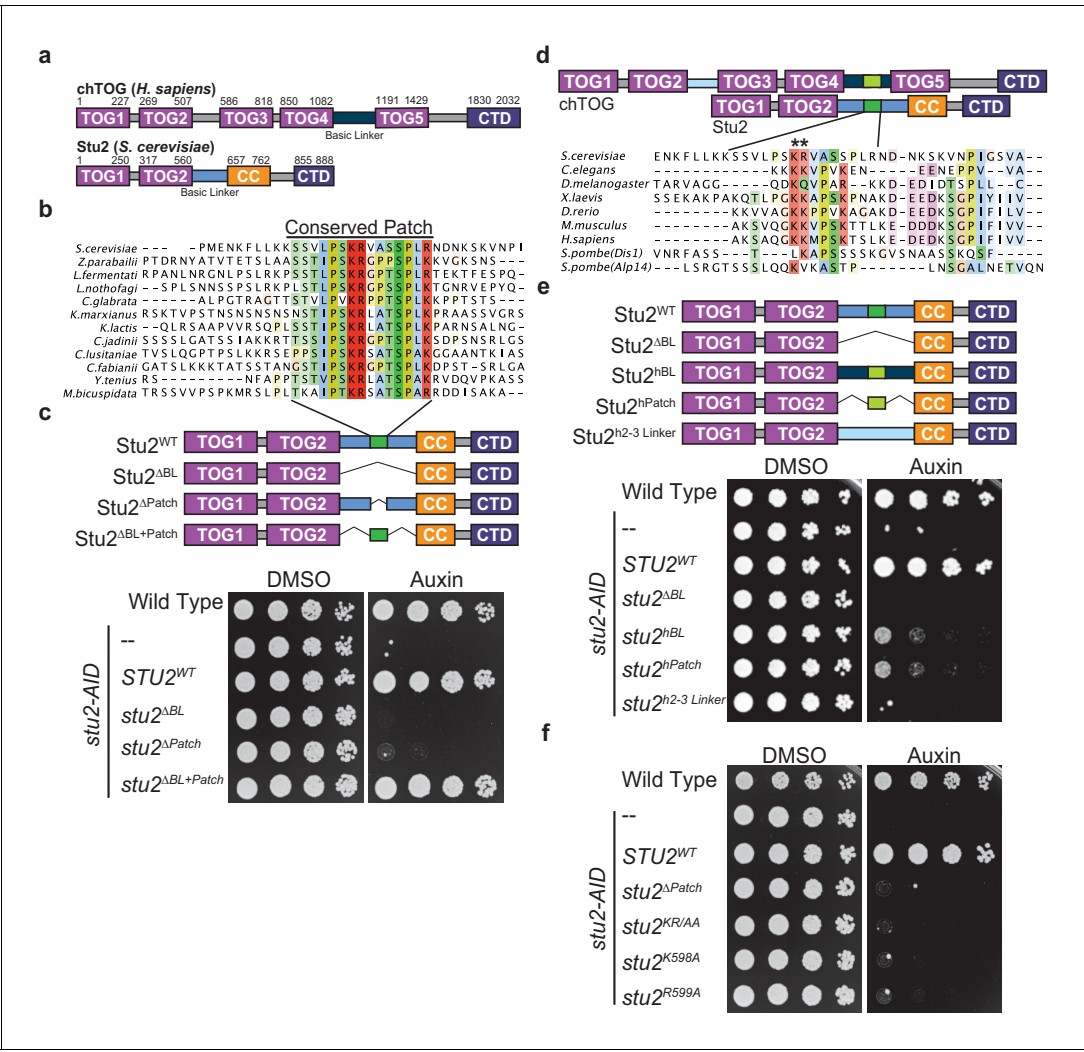

**Figure 2.** Two conserved basic residues are essential for Stu2 function. (**a**) Schematic of the yeast Stu2 and human chTOG proteins describing the domains in each protein. Specific residues marking domains are indicated on the top of each protein. (**b**) ClustalO multiple sequence alignment generated from full-length *Saccharomyces cerevisiae* Stu2 and related proteins in Ascomycota. Fifteen amino acids within the otherwise divergent ~110 amino acid 'basic linker' are colored based on percent conservation and biochemical properties of the side chain. (**c**) Schematic of the Stu2 mutant proteins used to investigate the essential nature of the conserved patch in *S. cerevisiae*. Cell viability was analyzed in *stu2-AID* strains expressing the indicated Stu2 mutant proteins by plating fivefold serial dilutions in the presence of vehicle (left) and or auxin (right) to degrade the endogenous Stu2-AID protein. (**d**) Schematic of human chTOG and yeast Stu2 proteins showing how ClustalO multiple sequence alignments were performed specifically on basic linker regions (below) from metazoan and fungal species (colored similarly to (**b**)). (**e**) Schematic of the Stu2 mutant proteins used to investigate the orthologous behavior of the human basic linker and conserved patch (linkers are colored to match (**d**)). Cell viability determined as in (**c**) using a serial dilution growth assay. (**f**) The pair of conserved basic residues identified by asterisks in (**d**) were mutated individually or as a pair to alanine and found to be required for *S. cerevisiae* viability as assayed in (**c**).

tools to identify mutants that potentially inactivate its kinetochore function. Previous cross-linking mass-spectrometry with yeast proteins revealed that both the Stu2 basic linker and C-terminus interact with the Ndc80 complex, but only the C-terminus was required for kinetochore association (*Miller et al., 2019*; *Zahm et al., 2020*). This suggested the basic linker may instead interact with the Ndc80 complex to regulate kinetochore-microtubule attachments.

To identify a putative functional motif in the basic linker, we aligned Stu2 orthologs from related Ascomycota and found that in addition to the previously described compositional positive charge bias, there was a conserved 15 amino acid sequence in the basic linker (*Figure 2b*). To understand the role of this previously unidentified conserved patch, we ectopically expressed various *stu2* mutants under their native promoter in a strain where the endogenous allele was fused to an auxin-

inducible degron (*stu2-AID*). With serial dilution growth assays, we analyzed the viability of the various mutant cells in the presence of auxin, which degrades the endogenous Stu2-AID protein (*Figure 2c*). These complementation studies confirmed that the basic linker is essential (*stu2*$^{\Delta BL}$) and further demonstrated that the conserved patch is also required for budding yeast viability (*stu2*$^{\Delta Patch}$) (*Miller et al., 2019*). Strikingly, this patch (flanked by small flexible peptides) was sufficient to replace the entire 98 residue basic linker (*stu2*$^{\Delta BL+Patch}$), despite a 75% reduction in total length and an 83% reduction in positive residues (*Figure 2c*). To determine if this patch is conserved from yeast to primates, we aligned the basic linkers among commonly studied Ophistokonts and found the sequence of this patch was poorly conserved outside Ascomycota and appeared to be centered on a pair of basic residues (*Figure 2d*). We tested if the region identified in the human protein was functionally orthologous to the conserved patch in Ascomycota by generating chimeric Stu2 proteins (*Figure 2e*). To start, we replaced the yeast basic linker with the orthologous human sequence (*stu2*$^{hBL}$) and found it partially complemented the loss of endogenous Stu2 (*Figure 2e*). This partial rescue could be attributed to the non-specific positive charge of this region as has been previously suggested, or to the presence of a conserved sequence motif. We therefore tested whether the 14 amino acids from chTOG that aligned to the Stu2 conserved patch were sufficient for the partial rescue of yeast cell viability (*Stu2*$^{hPatch}$). Cells expressing this chimeric protein (*stu2*$^{hPatch}$) were as proliferative as those containing the entire basic linker from the human protein (*Figure 2e*), suggesting sequence elements in this small region are more important than the overall net charge of the basic linker. To further test the role of net positive charge, we replaced the yeast basic linker with the sequence between TOG2 and TOG3 in chTOG (*Stu2*$^{h2-3\ Linker}$). This region and the basic linker between TOG4 and TOG5 have similar lengths, propensities for disorder, and isoelectric points, yet this chimera did not rescue yeast viability (*Figure 2e*). Taken together, these data suggest that the region we identified in the basic linker of chTOG is orthologous to the one we found in yeast Stu2.

Our discovery that a 15 amino acid conserved patch is sufficient for viability strongly suggested that the sequence of the region is more important than overall charge. We found a pair of basic residues in this region in both the yeast and human proteins, so we tested their function. We mutated Stu2 K598 and R599 to alanine as a pair (*stu2*$^{KR/AA}$) or separately and found that despite normal mutant protein levels (data not shown), the loss of either residue was lethal to yeast (*Figure 2f*). These data show that two conserved residues, rather than the net charge of the basic linker, are essential for cell viability. While the in vitro microtubule polymerase activity is regulated by net charge (*Geyer et al., 2018*; *Widlund et al., 2011*), our data suggest that in vivo there is a different essential role for the basic linker that is mediated by a small patch in this region.

We set out to determine whether these conserved residues are required for chTOG function in human cells. Past studies of chTOG have primarily focused on its binding partners or RNAi depletion phenotypes due to its large size and multiple cellular functions (*Gutiérrez-Caballero et al., 2015*; *Hood et al., 2013*). In addition, the constitutive overexpression of chTOG is toxic, making it difficult to study in mammalian cells. To overcome these technical challenges, we generated HeLa cell lines that harbor doxycycline-inducible, siRNA-resistant alleles of chTOG-EGFP (chTOG$^{WT}$) or chTOG [K1142, 1143A]-EGFP (chTOG$^{KK/AA}$ or basic pair mutant) (*Gossen and Bujard, 1992*; *O'Gorman et al., 1991*; *Taylor and McKeon, 1997*; *Figure 3a,b*). In these cells, chTOG siRNA functioned in a dose-dependent manner allowing for partial or near-complete depletion (*Figure 3—figure supplement 1a*). Doxycycline treatment of depleted cells resulted in equivalent levels of chTOG$^{WT}$ and chTOG$^{KK/AA}$ proteins, indicating that the mutations do not alter protein stability (*Figure 3—figure supplement 1a*).

To determine whether the basic pair mutant could support cell growth, we analyzed cell proliferation after depletion of endogenous protein and induction of the ectopic chTOG proteins. Expression of the WT chTOG protein restored viability, indicating that there were no major off-target siRNA effects (*Figure 3c*). In contrast, the chTOG basic pair mutant was not able to support cell proliferation after depletion of endogenous chTOG, despite expressing similarly to WT chTOG (*Figure 3c*). The failure to proliferate after chTOG depletion was previously shown to arise from a mitotic delay and resulting chromosome segregation errors (*Cassimeris and Morabito, 2004*; *Gergely et al., 2003*). To determine if chTOG$^{KK/AA}$ induced similar defects, we stained DNA and determined both the mitotic index and incidence of chromosome segregation errors. After chTOG depletion, we observed a fourfold increase in mitotic cells (20% of the population) as expected, which was significantly reduced when chTOG$^{WT}$ was expressed, but not when chTOG$^{KK/AA}$ was expressed

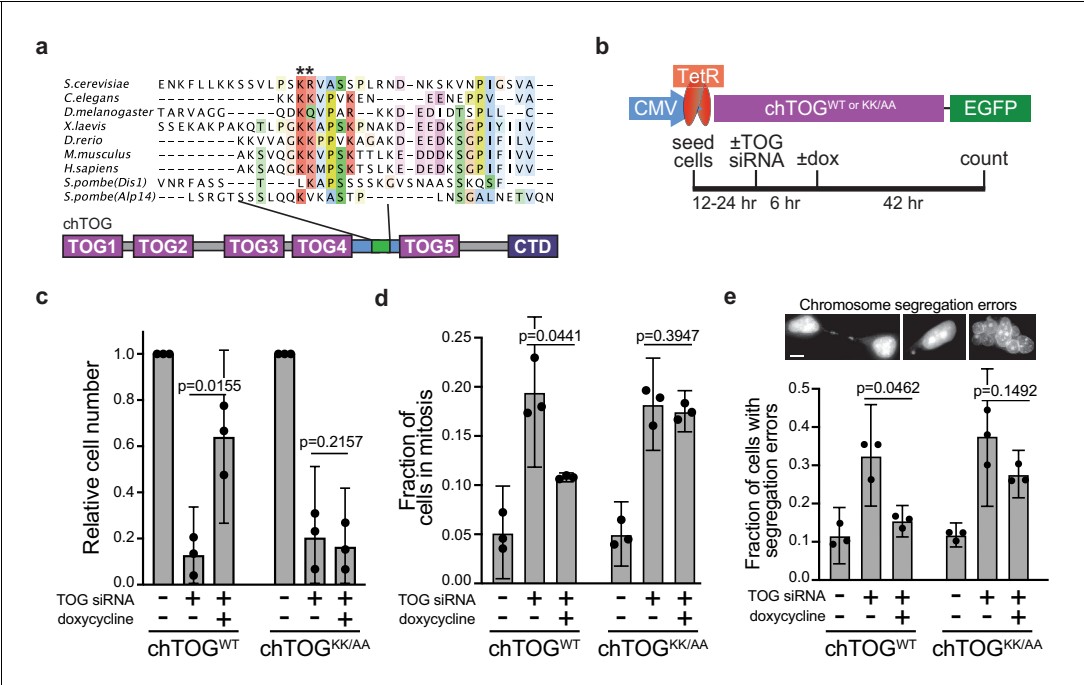

**Figure 3.** Mutating two conserved basic residues in chTOG causes mitotic delay and lethal chromosome segregation errors. (**a**) ClustalO multiple sequence alignment generated from basic linker regions of metazoan and fungal species shown in *Figure 2*. The pair of basic residues mutated to alanine denoted with asterisks. (**b**) Assay to analyze the ability of doxycycline-inducible codon-optimized, chTOG[WT] and chTOG[KK/AA] to complement cellular activities after siRNA-mediated depletion of endogenous chTOG. Cells depleted of chTOG or expressing ectopic chTOG proteins were assayed for (**c**) proliferation, (**d**) mitotic index, and (**e**) chromosome segregation errors. Scale bar is 5 µm; mean values and 95% confidence intervals for three experimental replicates displayed; p-values determined by paired t test.

The online version of this article includes the following figure supplement(s) for figure 3:

**Figure supplement 1.** chTOG[WT] and chTOG[KK/AA] are expressed at equivalent levels; mitotic delays from chTOG depletion or mutation require Mps1 activity.

(*Figure 3d*). The mitotic delays observed in chTOG-depleted and chTOG[KK/AA]-expressing cells were dependent on Mps1 kinase activity (*Figure 3—figure supplement 1b*), suggesting that they are the result of erroneous kinetochore-microtubule attachments that appropriately trigger the spindle assembly checkpoint. Consistent with this interpretation, the populations enriched for mitotic cells also displayed an increased incidence of chromosome segregation errors like anaphase bridges and micro or multiple nuclei (*Figure 3e*). These data suggest that chTOG depletion or mutation leads to defects in kinetochore-microtubule attachments that trigger the spindle assembly checkpoint, yet cells eventually exit mitosis in the presence of these defects, causing lethal chromosome segregation errors.

## Mutating the basic pair does not alter dynamics or structure of the microtubule cytoskeleton

The apparent kinetochore-microtubule attachment defects giving rise to observed chromosome segregation errors could arise from defects in a number of chTOG's activities including: (i) regulating cytoskeletal dynamics by nucleating and polymerizing microtubules, (ii) organizing the mitotic spindle into a bipolar structure, or (iii) regulating kinetochore microtubule binding activity. To understand if the basic linker contributed to these activities, we first analyzed the mutant by live cell TIRF microscopy on adherent interphase cells expressing EB1-mCherry (*Tinevez et al., 2017*; *Tirnauer et al., 2002*). chTOG requires functional TOG domains to bind microtubule plus-ends and elongate the polymer (*Widlund et al., 2011*), thus we verified that mutating the basic linker did not prevent the protein from localizing to growing plus-ends near EB1 (*Figure 4—figure supplement 1a*). To determine whether microtubule assembly rates were affected, we next measured the velocity

of EB1 comets after depletion of chTOG and observed an increase in microtubule assembly rates that was similarly suppressed by the expression of either chTOG[WT] or chTOG[KK/AA] in siRNA-treated cells (*Figure 4a*). Although this differs from other studies that reported either normal or decreased microtubule assembly rates upon chTOG knock down (*Cassimeris et al., 2009*; *Ertych et al., 2014*; *van der Vaart et al., 2011*), these discrepancies are likely due to technical differences (Appendix 1). Regardless, expression of the WT and basic pair mutant chTOG proteins restored microtubule dynamics equivalently (*Figure 4a*), suggesting that the essential function of the basic pair mutant is not related to regulating microtubule dynamics. chTOG depletion also leads to multipolar spindle formation (*Cassimeris and Morabito, 2004*; *Gergely et al., 2003*), so we next analyzed spindle morphology. As expected, nearly 50% of mitotic cells had multipolar spindles after chTOG depletion (*Figure 4b*). However, expression of the WT and basic pair mutant proteins fully supported bipolar spindle formation when the endogenous chTOG protein was depleted (*Figure 4b*). Thus, the basic pair mutant does not have detectable defects in the reported microtubule cytoskeleton functions of chTOG in vivo despite being essential for cell proliferation.

## chTOG destabilizes incorrect kinetochore-microtubule attachments

Microtubule dynamics and spindle formation were not affected by mutating the basic pair; thus, we tested the possibility that chTOG functions like the yeast ortholog in regulating kinetochore-

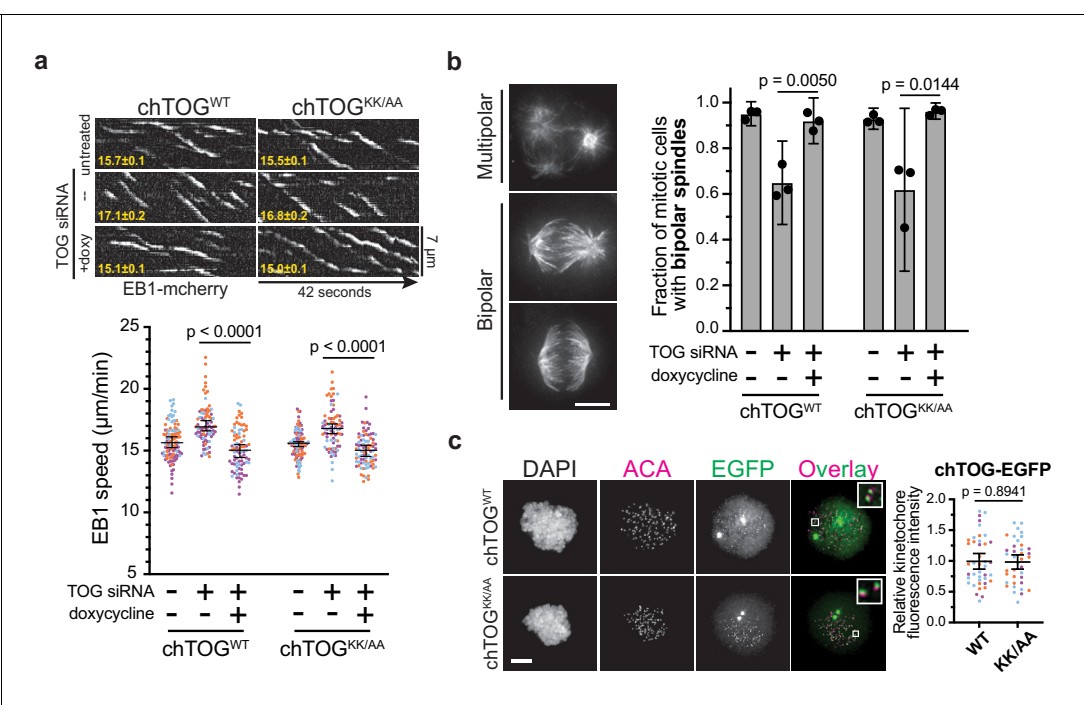

**Figure 4.** The chTOG basic pair mutant regulates microtubule dynamics, spindle structure, and localizes to kinetochores. (**a**) Images isolated from live-cell TIRF microscopy were used to generate kymographs of EB1-mCherry in chTOG[WT] and chTOG[KK/AA]-expressing cells with average EB1 track speed (μm/min) in bottom left and quantifications shown below. Each data point represents the mean EB1 track speed per cell and is grouped from three experimental replicates and colored according to each replicate with median and 95% confidence intervals displayed in black. p-Values were determined with an unpaired Mann-Whitney test. (**b**) Representative images of each spindle phenotype observed in mitotic chTOG-depleted, chTOG[WT], or chTOG[KK/AA] expressing cells. While bipolar spindles exhibited two distinct phenotypes, we first quantified the fraction of cells exhibiting multipolar or bipolar spindles. Mean values and 95% confidence interval for three replicates were reported, p values calculated with a paired t test. (**c**) Representative images (left) and quantification (right) of chTOG[WT] and chTOG[KK/AA] localization to kinetochores in the absence of endogenous chTOG. Cells were treated with nocodazole to eliminate microtubules. Each data point represents the mean chTOG-EGFP fluorescence intensity at all kinetochores in a single cell normalized to the mean value of chTOG[WT]-expressing cells. Data are grouped from three experimental replicates and colored according to each replicate with mean values and 95% confidence displayed in black. p-Value was determined with an unpaired t test. All scale bars are 5 μm.

The online version of this article includes the following figure supplement(s) for figure 4:

**Figure supplement 1.** The basic pair mutant localizes to microtubule plus-ends and interacts with Hec1.

microtubule attachments. First, we tested whether the basic pair mutant affected chTOG localization to the kinetochore using quantitative fluorescence microscopy and biochemical analysis in nocodazole-treated cells (*Figure 4c*, *Figure 4—figure supplement 1b,c*). There was no change in localization, consistent with our previous findings in yeast that the C-terminus of Stu2 is necessary and sufficient for stable association with the Ndc80 complex and kinetochores (*Miller et al., 2019*; *Zahm et al., 2020*). Thus, defects in the regulation of kinetochore-microtubule attachments in cells expressing the basic pair mutant are not due to altered protein localization.

Further phenotypic analysis of mitotic cells expressing the basic pair mutant showed a defect in chromosome alignment, where 90% of cells formed a poorly organized metaphase plate (*Figure 5a*). Most of the unaligned chromosomes were clustered at the poles with an excess of astral microtubules where they appeared to form stable syntelic or monotelic attachments (*Figure 5a*, image inlays). These attachments were reminiscent of Aurora B kinase inhibition, suggesting that kinetochore-microtubule attachments were prematurely stabilized, allowing errors to persist and preventing chromosome alignment (*Hauf et al., 2003*; *Kallio et al., 2002*).

To better characterize the kinetochore-microtubule attachment state of chTOG$^{KK/AA}$ expressing cells with unaligned chromosomes, we quantified the number of kinetochores with Mad1 staining because it specifically localizes to unattached kinetochores (*Hoffman et al., 2001*; *Howell et al., 2004*). Because unperturbed prometaphase cells have many unaligned chromosomes, there is high error correction activity that destabilizes these erroneous attachments and generates an average of 36 Mad1-positive kinetochores (*Figure 5b*). In contrast, chTOG-depleted cells with unaligned chromosomes only average 13 Mad1-positive kinetochores, and this can be rescued by expression of WT chTOG but not the basic pair mutant (*Figure 5b*). To ensure that this assay was reflecting attachment state rather than a defect in Mad1 recruitment, we treated chTOG$^{WT}$ and chTOG$^{KK/AA}$-expressing cells with nocodazole and found that ~ 50 kinetochores were Mad1 positive in both cell types (*Figure 5—figure supplement 1a*). These data demonstrated that more kinetochores were attached to microtubules after chTOG depletion or mutation, which could arise from attachments being prematurely stabilized. To directly test this, we asked if erroneously attached microtubules were resistant to cold-induced depolymerization. Cells were incubated in ice-cold growth medium for 8 min prior to fixation and immunostaining, then cells with bipolar spindles but unaligned chromosomes were imaged. This revealed that in untreated cells, kinetochores on the astral side of the mitotic spindle rarely formed cold-stable attachments (*Figure 5c*). However, in chTOG-depleted cells or those expressing the basic pair mutant, many of the erroneously attached kinetochores on the astral side of the spindle remained attached to microtubules after cold treatment (*Figure 5c*). We confirmed the stability of these microtubules by quantifying the fluorescence intensity of a region on the astral side of the spindle that encompassed all kinetochores and subtracting the background signal. These data suggest chTOG *destabilizes erroneous kinetochore-microtubule attachments*, and erroneous attachments persist when it is depleted or mutated. This would result in kinetochores bound to microtubules but lacking tension, which is consistent with our observation of a mitotic delay but eventual exit with chromosome segregation errors.

To further test if chTOG is required to correct low-tension, syntelic attachments, we treated cells with a reversible Eg5/KIF11 inhibitor (STLC) to arrest them in a monopolar state, which enriches for these attachments (*Kapoor et al., 2000*). We then assayed the relative number of microtubules bound by each kinetochore (*Dudka et al., 2018*). After chTOG depletion, kinetochore attached fibers contained 1.5x more microtubules per kinetochore (*Figure 5d*, *Figure 5—figure supplement 1b*). Expression of WT chTOG reversed this phenotype, while attachments remained hyper-stable when the basic pair mutant was expressed, even in the presence of endogenous chTOG (*Figure 5—figure supplement 1c*). This suggests that erroneous kinetochore-microtubule attachments are stabilized in the absence of chTOG and its basic pair activity.

Because mutation or deletion of chTOG appeared to stabilize the erroneous attachments generated by STLC treatment, we next directly tested whether chTOG is required to *correct* errors. We washed out the STLC and assayed if cells recovered from the monopolar state and aligned chromosomes. Immediately after STLC removal, few cells had aligned chromosomes in any experimental condition (*Figure 6—figure supplement 1a*). 60 min after STLC washout, a majority of control cells (not treated with siRNA) had formed a bipolar spindle and aligned chromosomes at the spindle equator (*Figure 6a*). As expected, chTOG-depleted cells failed to form bipolar spindles and therefore did not align chromosomes at a metaphase plate (*Figure 6a*, *Figure 6—figure supplement 1b*;

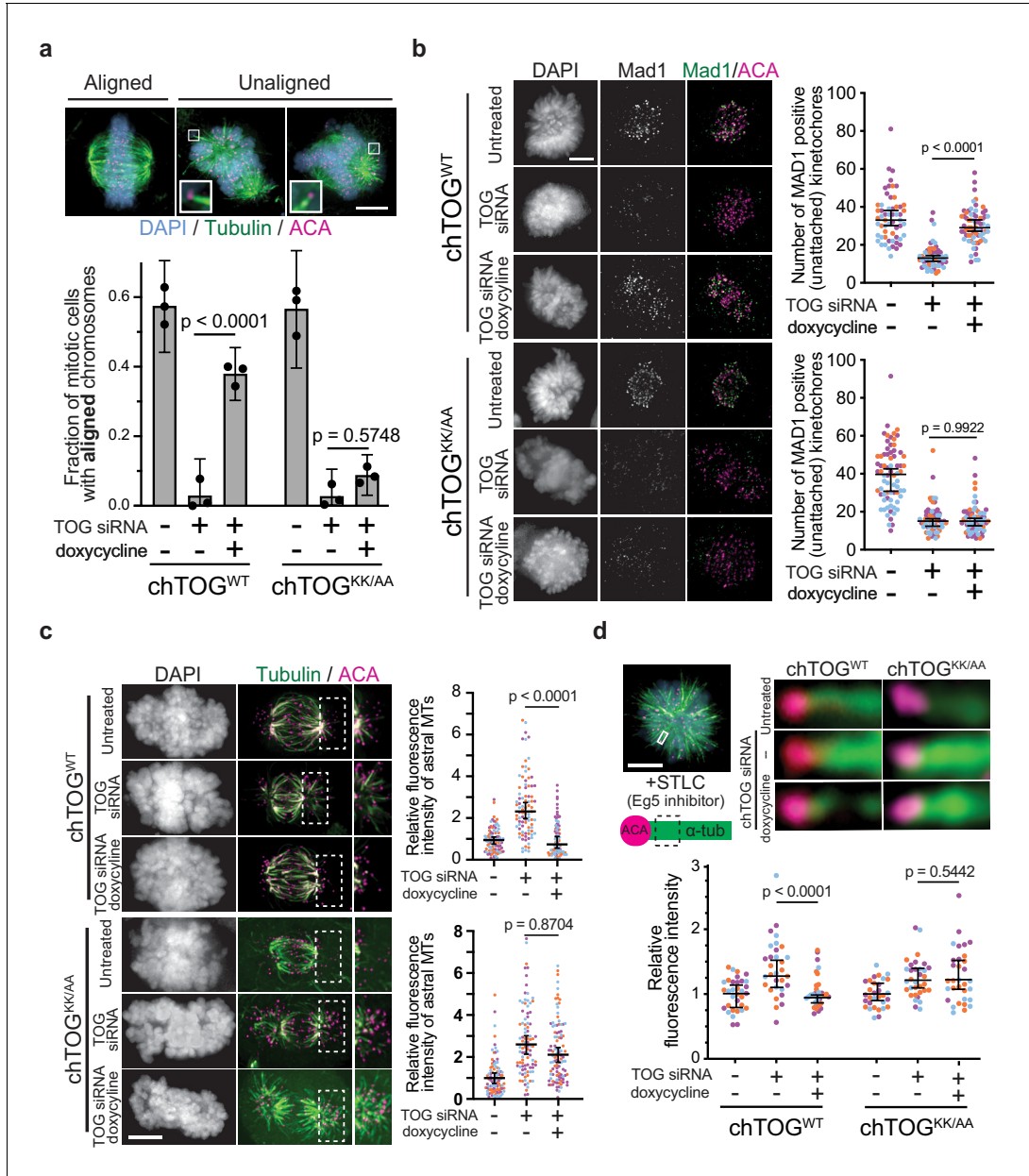

**Figure 5.** The chTOG basic pair is required to regulate kinetochore-microtubule attachment stability. (a) Representative images of each chromosome alignment phenotype observed in mitotic chTOG-depleted, chTOG$^{WT}$, or chTOG$^{KK/AA}$ show a large fraction of chTOG$^{KK/AA}$ expressing cells form bipolar spindles with excessive astral microtubules that attach to kinetochores (image inlays) and prevent chromosome alignment. Phenotypes are quantified below where mean values and 95% confidence intervals for three replicates are reported. p-Values were determined with a paired t test and contrast on inlays was adjusted independently. (b–d) Data points on graphs are grouped from three experimental replicates and colored according to each replicate with median and 95% confidence intervals displayed in black. p-Values determined by an unpaired Mann-Whitney test. (b) Representative images (left) and quantification (right) of Mad1 immunostaining as a marker for kinetochore-microtubule attachment state in chTOG depleted, chTOG$^{WT}$ (top), or chTOG$^{KK/AA}$ (bottom) expressing cells. Each data point represents the number of kinetochores with Mad1 puncta per cell. (c) Representative images (left) and quantification (right) of cold-stable astral (erroneous) kinetochore-microtubule attachments in chTOG depleted, chTOG$^{WT}$ (top), or chTOG$^{KK/AA}$ (bottom) expressing cells. Each data point represents the tubulin fluorescence intensity of all astral microtubules on one half of the mitotic spindle. (d) Monopolar spindles (top left) were formed by inhibiting Eg5/KIF11 with STLC to allow the fluorescence intensity quantification of kinetochore-bound microtubule bundles at low-tension, erroneous attachments in control cells or chTOG-depleted cells expressing chTOG$^{WT}$ or chTOG$^{KK/AA}$ (right). Each data point is relative intensity normalized to the mean of uninduced, mock-depleted cells. All scale bars are 5 μm.

The online version of this article includes the following figure supplement(s) for figure 5:

**Figure supplement 1.** The basic pair mutant does not affect Mad1 recruitment to kinetochores and competes with endogenous chTOG.

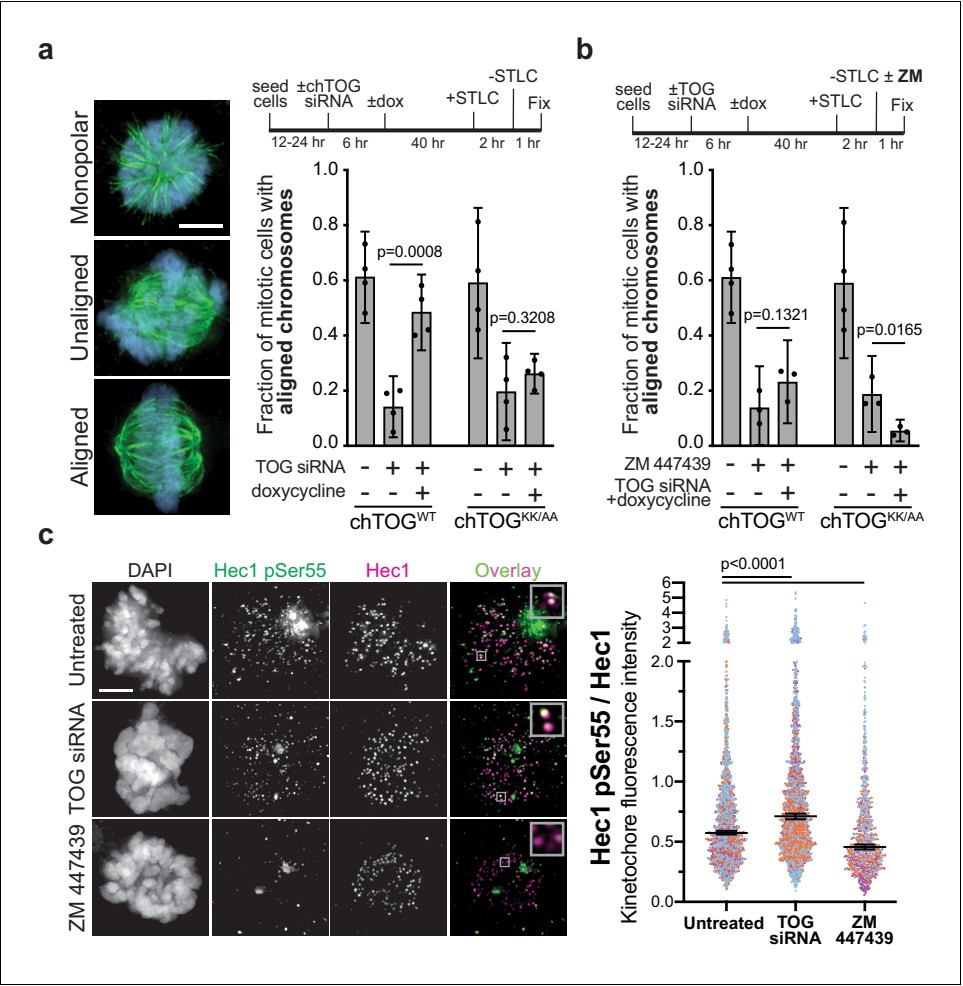

**Figure 6.** chTOG- and Aurora-B-dependent error pathways likely function independently. (a) Mitotic error correction was assayed by inducing errors with STLC to inhibit Eg5 and then washing out the inhibitor in control cells or chTOG-depleted cells expressing chTOG^WT or chTOG^KK/AA. The chromosome alignment phenotype (left) was quantified 60 min after inhibitor washout. (b) The same error correction assay was performed as in (a) but was supplemented with a low dose of Aurora B kinase inhibitor ZM 447439. Untreated populations in (a and b) are the same and both display mean values and the 95% confidence interval of three or four experimental replicates; p values were determined with paired t tests. (c) Representative immunofluorescence images (left) and quantifications (right) of the relative fluorescence intensity of phosphorylated Hec1 analyzed with a phospho-specific antibody to Ser55. Each data point represents individual kinetochore intensities of Hec1 pSer55 antibody ratioed to Hec1 antibody. Data points on graphs are grouped from three experimental replicates and colored according to each replicate with median and 95% confidence intervals displayed in black. p-Values were determined by an unpaired Mann-Whitney test. All scale bars are 5 µm.

The online version of this article includes the following figure supplement(s) for figure 6:

**Figure supplement 1.** Formation of bipolar or monopolar spindles in STLC washout experiments.

*Cassimeris and Morabito, 2004*; *Gergely et al., 2003*). In contrast, expression of either chTOG^WT or chTOG^KK/AA after siRNA treatment rescued bipolar spindle formation after 60 min; however, only chTOG^WT-expressing cells properly aligned their chromosomes (*Figure 6a*). Together, these data reveal that chTOG functions as a mitotic error correction factor, and this activity appears independent of its well-characterized role as a regulator of the microtubule cytoskeleton.

# Aurora B kinase phosphoregulation of Hec1 cannot compensate for loss of chTOG activity

Our work suggested that chTOG functions similarly to the budding yeast ortholog (Stu2) that confers an intrinsic tension-dependent microtubule binding behavior to kinetochores that is independent of the extrinsic signaling through the Aurora B pathway (*Miller et al., 2016*; *Miller et al., 2019*). To test this in human cells, we analyzed recovery from STLC when both pathways were inhibited. Because the complete inhibition of Aurora B with ZM447439 (ZM) prevents cells from forming bipolar spindles after STLC washout (*DeLuca et al., 2011*), we partially inhibited Aurora B with a lower dose (500 nM). Cells expressing WT chTOG, while Aurora B was inhibited, exhibited the same phenotype as ZM treatment alone. However, when the basic pair mutant was expressed in the presence of ZM, we observed an additive phenotype where essentially no cells formed an organized metaphase plate after 60 min (*Figure 6b*) despite the ability to form a bipolar spindle (*Figure 6—figure supplement 1b*). These data suggested that chTOG and Aurora B pathways function through separate mechanisms.

While chTOG and Aurora B appear to act on kinetochores through unique mechanisms, they both respond to the same signal of low-tension kinetochore-microtubule attachments. This leads to the question of why cells lacking chTOG or expressing the basic pair mutant can accumulate low-tension kinetochore-microtubule attachments (*Figure 5*). We wondered whether these errors were unable to trigger Aurora B activity, or whether Aurora B was activated but insufficient to destabilize these attachments. To examine this, we measured the phosphorylation status of the Aurora B substrate Hec1 upon chTOG depletion. We assayed cells containing unaligned chromosomes and found that chTOG depletion causes an increase in the phosphorylation of Ser55 on Hec1 (*Figure 6c*). We confirmed that the signal depends on Aurora B activity by partially inhibiting it and observing an expected ~50% decrease in signal (*Figure 6c*). The increased phosphorylation in the absence of chTOG only occurred at 10% of kinetochores in each cell (likely those stable erroneous attachments on the astral side of the spindle), making it imperceptible if kinetochore signals were averaged on a per cell basis. Together, these data suggest that Aurora B is activated when the chTOG error correction pathway is defective, but that it cannot fully compensate to destabilize these attachments.

## Discussion

Here, we have identified a previously unknown mitotic error correction pathway in human cells, where chTOG localizes to kinetochores and destabilizes erroneous, low-tension attachments. Without this activity, erroneous low-tension attachments persist and appropriately activate the spindle assembly checkpoint to generate a mitotic arrest. However, attached but low-tension kinetochores cannot maintain the checkpoint indefinitely and cells exit mitosis in the presence of kinetochore-microtubule attachment errors, resulting in lethal chromosome segregation defects. While numerous proteins have been implicated in selectively stabilizing bioriented attachments (*Gaitanos et al., 2009*; *Girão et al., 2020*; *Janke et al., 2002*; *Maiato et al., 2003*; *Ortiz et al., 2009*; *Raaijmakers et al., 2009*; *Schmidt et al., 2010*; *Welburn et al., 2009*), less is known about the mechanisms that recognize and destabilize errors. This activity of chTOG was likely not previously elucidated due to the pleiotropic effects of chTOG depletion on spindle structure and dynamics. By arresting cells with monopolar spindles and identifying a targeted mutation of the basic linker, we were able to specifically assay chTOG error correction functions independent of its regulation of the cytoskeleton.

Although the basic linker regulates polymerase activity in vitro via non-specific electrostatic interactions with the microtubule lattice (*Geyer et al., 2018*; *Wang and Huffaker, 1997*; *Widlund et al., 2011*), yeast viability was normal when the entire basic linker (pI = 10.4; 18 positive residues) was replaced with just 15 conserved residues (pI = 8.7; three positive residues). Most strikingly, yeast proliferation was partially maintained when the Stu2 basic linker was replaced with the chTOG basic linker (pI = 9.7), but not maintained when replaced with the chTOG linker between TOG2 and TOG3 (pI = 10.1). Moreover, mutation of one or two positive residue rendered yeast and human cells completely inviable. Therefore, while the basic linker net charge is a determinate for in vitro polymerase activity (*Geyer et al., 2018*; *Widlund et al., 2011*), it does not correlate with cell viability. We therefore propose that the essential function of the basic linker is to mediate error correction rather than regulate microtubule dynamics.

While the basic linker of many TOG orthologs contributes to microtubule lattice binding in vitro, it has also been proposed that the most C-terminal TOG domain in metazoan TOG proteins evolved to interact with the microtubule lattice rather than tubulin dimers (*Byrnes and Slep, 2017*). In particular, the crystal structure of TOG5 from the *Drosophila melanogaster* TOG protein, Msps, revealed an extra helix pair at its N-terminus that could make contacts with a lateral tubulin dimer within the microtubule lattice (*Byrnes and Slep, 2017*). The basic pair is 20 amino acids upstream of this unique helix pair, making it possible that they may contribute to TOG5 lattice binding activity. However, mutation of the unique N-terminal helices or the canonical tubulin binding residues in TOG5 caused defects in microtubule dynamics and spindle organization (*Byrnes and Slep, 2017*), while the basic pair mutant does not appear to compromise any of these activities. Thus, the basic pair is unlikely to contribute to TOG5 lattice binding, but in the future it will be important to determine if and how the tubulin-/microtubule-binding capacity of each TOG domain contributes to mitotic error correction.

We also found that the conserved process of chTOG-based error correction is most likely independent of Aurora B phosphorylation of Hec1, similar to budding yeast (*Miller et al., 2019*). In fact, our data suggest that Aurora B is active at the low-tension attachments observed when chTOG error correction is inhibited, yet this phosphorylation is not sufficient to compensate for loss of chTOG-mediated error correction. Thus, we envision two error correction pathways that respond to overlapping input signals but leverage unique molecular strategies to destabilize erroneous kinetochore-microtubule attachments. Currently, it is difficult to understand why cells would utilize multiple non-

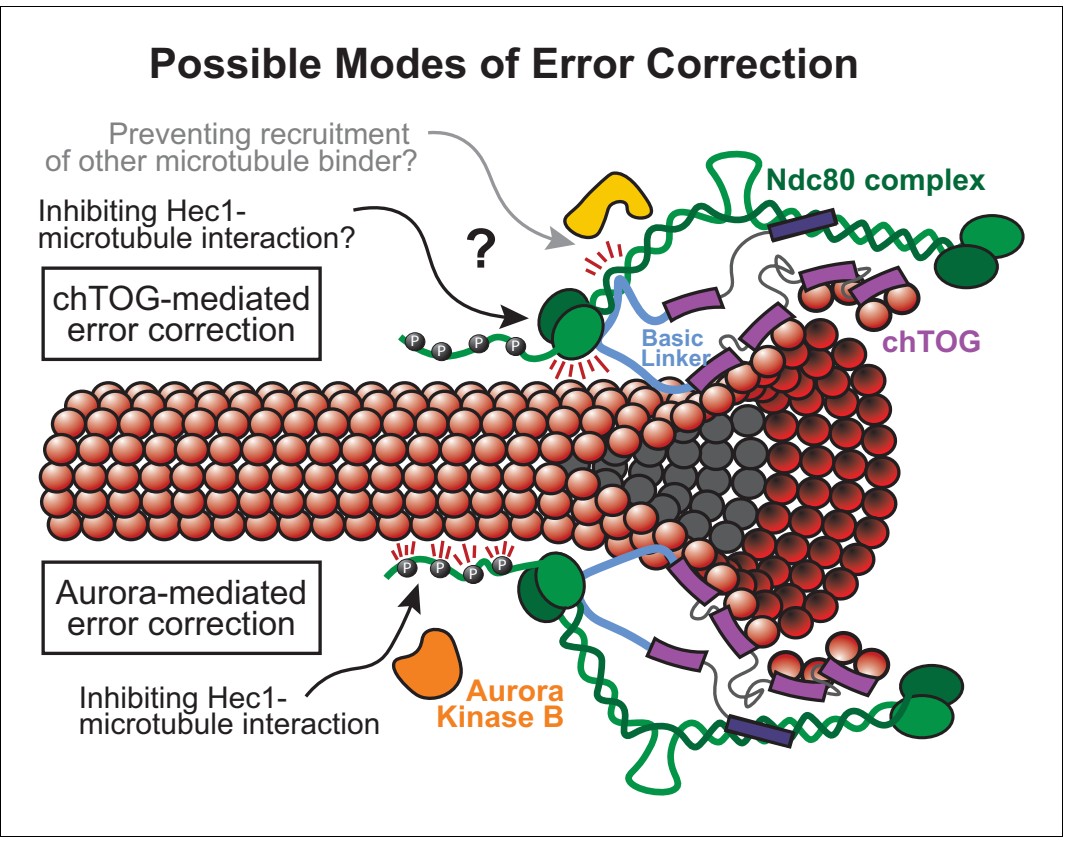

**Figure 7.** Possible model for chTOG-mediated error correction, independent of Aurora B kinase activity. Possible models of chTOG-mediated error correction (top) where chTOG kinetochore localization depends on the C-terminus while the TOG domains are still capable of binding tubulin at the microtubule tip in the 'bent' conformation. This may position the basic linker near the Ndc80/Hec1 CH domain to regulate Ndc80 complex activity. We favor the basic linker directly modulating Hec1 microtubule binding behavior, but chTOG could also prevent recruitment of other microtubule binding proteins. This activity is independent of Aurora-B-mediated error correction (bottom) that also recognizes low-tension attachments in the absence of chTOG, phosphorylates the Ndc80 tail domain, but on average cannot fully destabilize the bond.

**Table 1.** Plasmids used in this study.

| Plasmid | Vector backbone | Gene of interest | Mutation description | Selection marker | Primers | Source |
|---|---|---|---|---|---|---|
| pSB2232 | pSB2223/pL300 | Stu2$^{WT}$-3V5 | | LEU2 | SB4372, SB4374 | *Miller et al., 2016* PMID:27156448 |
| pSB2260 | pSB2223/pL300 | Stu2$^{\Delta BL}$-3V5 | $\Delta$560–657: : GDGAG | LEU2 | SB4411, SB4413 | *Miller et al., 2019* PMID:31584935 |
| pSB2634 | pSB2223/pL300 | Stu2$^{\Delta Patch}$-3V5 | $\Delta$592–607: : GDGAG | LEU2 | SB5248, SB4413 | This study |
| pSB2820 | pSB2223/pL300 | Cloning intermediate | $\Delta$569–657:: GDGAG+ 592–607+GDGAG | LEU2 | SB5447 | This study |
| pSB2852 | pSB2223/pL300 | Stu2$^{\Delta BL+Patch}$-3V5 | $\Delta$560–657:: GDGAG+ 592–607+GDGAG | LEU2 | SB5519, SB5520 | This study |
| pSB2818 | pSB2223/pL300 | Stu2$^{\Delta K598A}$-3V5 | K598A | LEU2 | SB5458, SB4413 | This study |
| pSB2819 | pSB2223/pL300 | Stu2$^{\Delta R599A}$-3V5 | R599A | LEU2 | SB5459, SB4413 | This study |
| pSB2781 | pSB2223/pL300 | Stu2$^{\Delta KR598AA}$-3V5 | K598A, R599A | LEU2 | SB5349, SB4413 | This study |
| pSB3076 | pSB2223/pL300 | Stu2$^{hBL}$-3V5 | $\Delta$560–657::chTOG$^{1081-1167}$ | LEU2 | SB5248, SB4413 | This study |
| pSB3075 | pSB2223/pL300 | Stu2$^{hPatch}$-3V5 | $\Delta$560–657::GDGG+ chTOG$^{1137-1150}$+GDGAG | LEU2 | SB5447 | This study |
| pSB3089 | pSB2223/pL300 | Stu2$^{h2-3\ Linker}$-3V5 | $\Delta$560–657::chTOG$^{500-585}$ | LEU2 | SB5519, SB5520 | This study |
| pSB2822 | pCDNA3.1 | chTOG$^{WT}$-EGFP | | Neomycin | N/A | This study |
| pSB2823 | pCDNA3.1 | chTOG$^{KK1142AA}$-EGFP | K1142A, K1143A | Neomycin | N/A | This study |
| pSB2353 | pCDNA5 FRT/TO | N/A | | Puromycin | N/A | *Etemad et al., 2015* PMID:26621779 |
| pSB2860 | pCDNA5 FRT/TO | chTOG$^{WT}$-EGFP | | Puromycin | SB5536, SB5537 | This Study |
| pSB2863 | pCDNA5 FRT/TO | chTOG$^{KK1142AA}$-EGFP | K1142A, K1143A | Puromycin | SB5536, SB5537 | This Study |
| pSB2976 | pCDNA5 FRT/TO | chTOG$^{WT}$-6His-3Flag | | Puromycin | SB5774, SB5775 | This Study |
| pSB2977 | pCDNA5 FRT/TO | chTOG$^{KK1142AA}$-6His-3Flag | K1142A, K1143A | Puromycin | SB5774, SB5775 | This Study |
| pSB3062 | | EB1-mCherry | | Neomycin | N/A | Davidson Lab (Addgene: 55035) |
| pSB2998 | pLPH2 | N/A | | Hygromycin | N/A | This Study |
| pSB3217 | pLPH2 | EB1-mCherry | | Hygromycin | SB5938, SB5939, SB5940, SB5941 | This Study |

redundant mechanisms of error correction. One possibility is that Aurora B and chTOG error correction normally function in a more redundant manner, but the aneuploid and transformed nature of HeLa cells has broken this redundancy. Despite the lack of redundancy, both pathways seem to converge their activities at chTOG's direct binding partner, the Ndc80 complex. While the molecular mechanism of chTOG error correction is still unknown, it is more likely to inhibit Hec1 microtubule binding activity than to directly modulate microtubules. We suggest this because the basic pair mutant retains the ability to bind to kinetochores and microtubule tips, yet it fails to destabilize erroneous kinetochore-microtubule attachments, similar to depletion of chTOG (*Figure 5c,d*). Thus, we suggest a model where the C-terminus of Stu2/chTOG is required for stable association with the kinetochore (*Miller et al., 2019*; *Zahm et al., 2020*) while the basic linker modulates Hec1's microtubule-binding behavior through competitive or allosteric means (*Figure 7*). While it is also possible the basic linker inhibits the recruitment of other microtubule-binding factors to the Nc80 complex, this is less likely because many of these factors are unique to budding yeast (Dam1/DASH complex) or metazoans (SKA and Astrin-SKAP complexes) (*van Hooff et al., 2017a*; *van Hooff et al., 2017b*), yet the basic linker functions similarly in both species. Understanding the relationship between the Ndc80 complex and the basic linker in the context of microtubule attachment will be a key area of future research. chTOG error correction activity also provides new perspectives for cancer biology.

RNAi screens for cancer vulnerabilities suggested that some tumors grow dependent or 'addicted' to elevated levels of chTOG, making it a therapeutic target. However, chTOG depletion is reminiscent of microtubule poisons that are universally toxic, but particularly potent in specific cancers (*Martens-de Kemp et al., 2013*; *Tiedemann et al., 2012*). Dose-limiting toxicities are a common challenge for developing anti-mitotic therapies, but preclinical studies indicate that they can be overcome through precision inhibition of single functions within multifunctional proteins (*Ding et al., 2013*). Future analyses of the basic pair mutant will reveal if specifically inhibiting chTOG-mediated error correction, rather than degrading the entire protein, is therapeutically advantageous.

# Materials and methods

## Key resources table

| Reagent type (species) or resource | Designation | Source or reference | Identifiers | Additional information |
|---|---|---|---|---|
| Chemical compound, drug | Nocodazole | Sigma-Aldrich | M1404 | |
| Chemical compound, drug | Thymidine | Sigma-Aldrich | T9250 | |
| Chemical compound, drug | S-trityl-L-cysteine (STLC) | Sigma-Aldrich | 164739 | |
| Chemical compound, drug | Reversine | Sigma-Aldrich | R3904 | |
| Chemical compound, drug | ZM447439 | Selleckchem | S1103 | |
| Chemical compound, drug | Puromycin Dihydrochloride | Sigma-Aldrich | A11138-03 | |
| Chemical compound, drug | Hygromycin B | Thermo Fisher | 10687010 | |
| Chemical compound, drug | Doxycycline | Sigma-Aldrich | D9891 | |
| Chemical compound, drug | Indole-3-acetic acid (Auxin) | Sigma-Aldrich | I3750 | |
| Chemical compound, drug | Polyethyleneimine (PEI) | Polysciences | 23966–1 | Linear, MW 25000 |
| Chemical compound, drug | Lipfectamine 2000 | Thermo Fisher | 11668027 | |
| Chemical compound, drug | Lipofectamine RNAiMax | Thermo Fisher | 13778075 | |
| Other | Protein G Dynabeads | Thermo Fisher | 10009D | |
| Other | DAPI stain | Thermo Fisher | D1306 | IF: 60 ng/mL |
| Antibody | Anti-Flag (M2) [mouse monoclonal] | Millipore-Sigma | Cat# F3165; RRID:AB_259529 | IP (1 µg / 60 µL Prot G bead) |
| Antibody | Anti-GAPDH (6C5) [mouse monoclonal] | Millipore-Sigma | Cat# MAB374; RRID:AB_2107445 | WB (1 µg/mL) |

*Continued on next page*

*Continued*

| Reagent type (species) or resource | Designation | Source or reference | Identifiers | Additional information |
|---|---|---|---|---|
| Antibody | Anti-CKAP5(chTOG) [rabbit polyclonal] | GeneTex | Cat# GTX30693; RRID:AB_625852 | WB (1:1000) |
| Antibody | Anti-Hec1 (9G3) [mouse monoclonal] | Thermo Fisher | Cat# MA1-23308; RRID:AB_2149871 | WB (2 µg/mL) IF (1 µg/mL) |
| Antibody | Anti-GFP (JL-8) [mouse monoclonal] | Takara | Cat# 632381; RRID:AB_2313808 | WB (0.5 µg/mL) |
| Antibody | HRP-conjugated anti-mouse [sheep polyclonal] | GE Healthcare | Cat# NA931; RRID:AB_772210 | WB (1:10,0000) |
| Antibody | HRP-conjugated anti-rabbit [sheep polyclonal] | GE Healthcare | Cat# NA934; RRID:AB_2722659 | WB (1:10,0000) |
| Antibody | Anti-centromere antibody (ACA) [human polyclonal] | Antibodies Inc | Cat# 15–235; RRID:AB_2797146 | IF (1:600) |
| Antibody | Anti-alpha tubulin (DM1A) [mouse monoclonal] | Millipore-Sigma | Cat# T6199; RRID:AB_477583 | IF (2 µg/mL) |
| Antibody | Anti-Mad1 [rabbit polyclonal] | GeneTex | Cat# GTX109519; RRID:AB_1950847 | IF (1:1000) |
| Antibody | Anti-pSer55 Hec1 [rabbit purified polyclonal] | *DeLuca et al., 2011* PMID:21266467 | | IF (1:1000) |
| Antibody | Alexa Fluor 594 conjugated anti-mouse [goat polyclonal] | Thermo Fisher | Cat# A11005; RRID:AB_2534073; | IF (1:300; 1:600 for Tubulin) |
| Antibody | Alexa Fluor 647 conjugated anti-mouse [goat polyclonal] | Thermo Fisher | Cat# A21235; RRID:AB_2535804 | IF (1:300; 1:600 for Tubulin) |
| Antibody | Alexa Fluor 594 conjugated anti-rabbit [goat polyclonal] | Thermo Fisher | Cat# A11037; RRID:AB_2534095 | IF (1:300) |
| Antibody | Alexa Fluor 647 conjugated anti-rabbit [goat polyclonal] | Thermo Fisher | Cat # A21244; RRID:AB_2535812 | IF (1:300) |
| Antibody | Alexa Fluor 594 conjugated anti-human [goat polyclonal] | Thermo Fisher | Cat# A11014; RRID:AB_2534081 | IF (1:300) |
| Antibody | AlexFluor 647 conjugated anti-human [goat polyclonal] | Thermo Fisher | Cat# A21445; RRID:AB_2535862 | IF (1:300) |
| Transfected construct | siRNA to Hec1 (custom sequence) | Qiagen | | CCCUGGGUCG UGUCAGGAA |
| Transfected construct | siRNA to chTOG (flexitube) | Qiagen | Cat# SI02653588 | AAGGGTCGAC TCAATGATTCA |
| Recombinant DNA reagent | Stu2$^{WT}$-3V5 | *Miller et al., 2016* PMID:27156448 | pSB2232 | See *Table 1* for more details |

*Continued on next page*

*Continued*

| Reagent type (species) or resource | Designation | Source or reference | Identifiers | Additional information |
|---|---|---|---|---|
| Recombinant DNA reagent | Stu2$^{\Delta BL}$-3V5 | *Miller et al., 2019* PMID:31584935 | pSB2260 | See *Table 1* for more details |
| Recombinant DNA reagent | Stu2$^{\Delta Patch}$-3V5 | This study | pSB2634 | See *Table 1* for more details |
| Recombinant DNA reagent | Cloning intermediate | This study | pSB2820 | See *Table 1* for more details |
| Recombinant DNA reagent | Stu2$^{\Delta BL+Patch}$-3V5 | This study | pSB2852 | See *Table 1* for more details |
| Recombinant DNA reagent | Stu2$^{\Delta K598A}$-3V5 | This study | pSB2818 | See *Table 1* for more details |
| Recombinant DNA reagent | Stu2$^{\Delta R599A}$-3V5 | This study | pSB2819 | See *Table 1* for more details |
| Recombinant DNA reagent | Stu2$^{\Delta KR598AA}$-3V5 | This study | pSB2781 | See *Table 1* for more details |
| Recombinant DNA reagent | Stu2$^{hBL}$-3V5 | This study | pSB3076 | See *Table 1* for more details |
| Recombinant DNA reagent | Stu2$^{hPatch}$-3V5 | This study | pSB3075 | See *Table 1* for more details |
| Recombinant DNA reagent | Stu2$^{h2-3\ Linker}$-3V5 | This study | pSB3089 | See *Table 1* for more details |
| Recombinant DNA reagent | pCDNA3_chTOG$^{WT}$-EGFP | This study | pSB2822 | See *Table 1* for more details |
| Recombinant DNA reagent | pCDNA3_chTOG$^{KK1142AA}$-EGFP | This study | pSB2823 | See *Table 1* for more details |
| Recombinant DNA reagent | FRT/TO | *Etemad et al., 2015* | pSB2353 | See *Table 1* for more details |
| Recombinant DNA reagent | FRT/TO_chTOG$^{WT}$-EGFP | This Study | pSB2860 | See *Table 1* for more details |
| Recombinant DNA reagent | FRT/TO_chTOG$^{KK1142AA}$-EGFP | This Study | pSB2863 | See *Table 1* for more details |
| Recombinant DNA reagent | FRT/TO_chTOG$^{WT}$-6His-3Flag | This Study | pSB2976 | See *Table 1* for more details |
| Recombinant DNA reagent | FRT/TO_chTOG$^{KK1142AA}$_6His-3Flag | This Study | pSB2977 | See *Table 1* for more details |
| Recombinant DNA reagent | EB1-mCherry | Addgene | RRID:Addgene_55035 | See *Table 1* for more details |
| Recombinant DNA reagent | pLPH2 | This Study | pSB2998 | See *Table 1* for more details |
| Recombinant DNA reagent | pLPH2_EB1-mCherry | This Study | pSB3217 | See *Table 1* for more details |
| Strain, strain background (*Saccharomyces cerevisiae*) | W303 | *Miller et al., 2016* PMID:27156448 | SBY3 | See *Table 2* for more details |
| Genetic reagent (*S. cerevisiae*) | STU2-IAA7; TIR1 | *Miller et al., 2016* PMID:27156448 | SBY13772 | See *Table 2* for more details |
| Genetic reagent (*S. cerevisiae*) | STU2-IAA7; TIR1; Stu2$^{WT}$ | *Miller et al., 2019* PMID:31584935 | SBY13901 | See *Table 2* for more details |
| Genetic reagent (*S. cerevisiae*) | STU2-IAA7; TIR1; Stu2$^{\Delta BL}$ | This study | SBY17069 | See *Table 2* for more details |

*Continued on next page*

*Continued*

| Reagent type (species) or resource | Designation | Source or reference | Identifiers | Additional information |
|---|---|---|---|---|
| Genetic reagent (*S. cerevisiae*) | STU2-IAA7; TIR1; Stu2$^{KR/AA}$ | This study | SBY17206 | See *Table 2* for more details |
| Genetic reagent (*S. cerevisiae*) | STU2-IAA7; TIR1; Stu2$^{K598A}$ | This study | SBY17477 | See *Table 2* for more details |
| Genetic reagent (*S. cerevisiae*) | STU2-IAA7; TIR1; Stu2$^{R599A}$ | This study | SBY17479 | See *Table 2* for more details |
| Genetic reagent (*S. cerevisiae*) | STU2-IAA7; TIR1; Stu2$^{\Delta Patch}$ | This study | SBY17519 | See *Table 2* for more details |
| Genetic reagent (*S. cerevisiae*) | STU2-IAA7; TIR1; Stu2$^{\Delta BL+Patch}$ | This study | SBY17593 | See *Table 2* for more details |
| Genetic reagent (*S. cerevisiae*) | STU2-IAA7; TIR1; Stu2$^{hBL}$ | This study | SBY18799 | See *Table 2* for more details |
| Genetic reagent (*S. cerevisiae*) | STU2-IAA7; TIR1; Stu2$^{hPatch}$ | This study | SBY18797 | See *Table 2* for more details |
| Genetic reagent (*S. cerevisiae*) | STU2-IAA7; TIR1; Stu2$^{h2-3\ Linker}$ | This study | SBY19023 | See *Table 2* for more details |
| Genetic reagent (*Homo sapiens*) | HCT116 chTOG-FKBP-EGFP | *Cherry et al., 2019* PMID:31058365 | SBM004 | See *Table 3* for more details |
| Cell line (*H. sapiens*) | HEK 293T | *Ding et al., 2013* PMID:23154965 | SBM033 | See *Table 3* for more details |
| Genetic reagent (*H. sapiens*) | HeLa FlpIn Trex | *Etemad et al., 2015* PMID:26621779 | SBM001 | See *Table 3* for more details |
| Genetic reagent (*H. sapiens*) | HeLa FlpIn Trex; chTOG$^{WT}$-EGFP | This study | SBM044 | See *Table 3* for more details |
| Genetic reagent (*H. sapiens*) | HeLa FlpIn Trex; chTOG$^{KK/AA}$-EGFP | This study | SBM046 | See *Table 3* for more details |
| Genetic reagent (*H. sapiens*) | HeLa FlpIn Trex; chTOG$^{WT}$-EGFP; EB1-mCherry | This study | SBM045 | See *Table 3* for more details |
| Genetic reagent (*H. sapiens*) | HeLa FlpIn Trex; chTOG$^{KK/AA}$-EGFP; EB1-mCherry | This study | SBM047 | See *Table 3* for more details |
| Gene (*S. cerevisiae*) | STU2 | Saccharomyces Genome Database | SGD:S000004035 | |
| Gene (*H. sapiens*) | chTOG; CKAP5 | Consensus Coding DNA Sequence Database | CCDS: 31477.1 | |
| Gene (*H. sapiens*) | EB1; MAPRE1 | Consensus Coding DNA Sequence Database | CCDS: 13208.1 | |
| Software, algorithm | Prism 9 | GraphPad Software | | Version 9.0.0 (86) |
| Software, algorithm | TrackMate | *Tinevez et al., 2017* PMID:27713081 | | Version 4.0.0 |

## Mammalian cell culture

HCT116 (*Cherry et al., 2019*), 293T (*Ding et al., 2013*), and HeLa FlpIn Cells (*Etemad et al., 2015*) cells were grown in a high-glucose DMEM (Thermo Fisher Scientific 11-965-118/Gibco 11965118) supplemented with antibiotic/antimycotic (Thermo Fisher Scientific 15240062) and 10% Foetal Bovine Serum (Thermo Fisher Scientific 26140095) at 37°C supplemented with 5% $CO_2$. For microscopy experiments, cell suspensions were added to 35 mm wells containing acid washed 1.5 × 22 mm square coverslips (Fisher Scientific 152222) and grown for 12–24 hr prior to transfections or immunostaining. For live-cell microscopy experiments, cell suspensions were added to 35-mm glass-

bottom dishes (Mattek Corp. P35G-1.5–20 C) and grown in standard growth media because they were performed in an environmental chamber using TIRF microscopy. Cell identity was not validated by STTR profiling, but each cell line contains unique genomic modifications (FRT recombination and zeomycin resistance, EGFP fusions) that were validated by expression of transgenes. Cell lines are regularly screened for mycoplasma contamination using DAPI staining.

To entirely depolymerize the microtubule cytoskeleton, growth media were supplemented with 10 µM nocodazole (Sigma-Aldrich M1404) for one hour. To synchronize cells, they were treated with 2.5 mM thymidine (Sigma-Aldrich T9250) for 16 hr, cells were then placed in drug-free media for 8 hr. A second synchronization was achieved by supplementing media with 2.5 mM thymidine again for 16 hr. Finally, thymidine was removed and 4–5 hr later (depending on cell line) 10 µM nocodazole was added. Eg5 inhibition was achieved by supplementing growth media with 5 µM S-trityl-L-cysteine (STLC, Sigma-Aldrich 164739) for 2 hr. Partial inhibition of Aurora B kinase was performed with a one-hour long treatment of 500 nM ZM447439 (Selleckchem S1103). Mps1 was inhibited by supplementing cell culture media with 2 µM Reversine (Sigma-Aldrich R3904).

## Immunoprecipitations

15 cm dishes of 293 T cells were transfected with 40 µg of empty vector (pSB2353), chTOG$^{WT}$-3Flag (pSB2976), or chTOG$^{KK/AA}$-3Flag (pSB2977) using 85 µg of polyethyleneimine (PEI, Polysciences 23966–1) as previously reported (*Longo et al., 2013*). Cells were harvested by mechanical dissociation with PBS and then centrifuged. The cell pellet was resuspended in 250 µL of complete lysis buffer (25 mM HEPES, 2 mM MgCl$_2$, 0.1 mM EDTA, 0.5 mM EGTA, 15% Glycerol, 0.1% NP-40, 150 mM KCl, 1 mM PMSF, 1 mM sodium pyrophosphate, 1x Pierce Protease Inhibitor Cocktail [Thermo Scientific 88666]) and snap frozen in liquid nitrogen. Samples were thawed and sonicated with a CL-18 microtip for 20 s at 50% maximum power with no pulsing three times using a Fisher Scientific FB50 sonicator. Approximately 150 U of Benzonase nuclease (Millipore E1014) was added to samples and incubated at room temperature for 5 min. The samples were centrifuged at 16,100 x g at 4° C in a tabletop centrifuge. Clarified lysates were moved to fresh microfuge tubes and 60 µL of Protein G Dynabeads (Thermo Fisher 10009D) conjugated with anti-FLAG(M2) monoclonal antibody (Sigma Aldrich F3165) as previously described (*Akiyoshi et al., 2009*) were added and incubated at 4°C with rotation for 90 min. Beads were washed four times with lysis buffer lacking PMSF, sodium pyrophosphate, and protease inhibitor cocktail. Proteins were eluted from beads in 40 µL of SDS sample buffer incubated at 95°C for 5 min.

## Immunoblotting

Clarified lysates prepared as described above were diluted in 2x SDS sample buffer and incubated at 95°C for 5 min. Immunoblots for chTOG were prepared by resolving lysates on NuPAGE 4–12% Bis-Tris Gels (Life Technologies, NP0329BOX) in 1x MOPS-SDS buffer and then transferring the proteins to 0.45 µm nitrocellulose membrane (BioRad, 1620115) for 2 hr at 4°C in transfer buffer containing 20% methanol. Membranes were washed in PBS+0.05% Tween-20 (PBS-T) and blocked with PBS-T+5% non-fat milk overnight at 4°C. Primary antibodies were diluted in PBS-T by the following factors or to the following concentrations: anti-GAPDH clone 6C5 (Millipore Sigma MAB374) 1 µg/mL; anti-CKAP5(chTOG) (GeneTex GTX30693) 1:1000; anti-HEC1 clone 9G3 (ThermoFisher Scientific, MA1-23308) 2 µg/mL; anti-GFP clone JL-8 (Takara 632381) 0.5 µg/mL. HRP-conjugated anti-mouse (GE Lifesciences, NA931) and anti-rabbit (GE Lifesciences, NA934) secondary antibodies were diluted 1:10,000 in PBS-T and incubated on membranes for 45 min at room temperature. Immunoblots were developed with enhanced chemiluminescence HRP substrates SuperSignal West Dura (Thermo Scientific, 34076) or SuperSignal West Femto (Thermo Scientific, 34094). All chemiluminescence was detected using a ChemiDoc MP system (BioRad).

## Immunofluorescent staining

Upon completion of experimental manipulations, cells grown on coverslips were immediately chemically crosslinked for 15 min with 4% PFA diluted from a 16% stock solution (Electron Microscopy Sciences, 15710) with 1x PHEM (60 mM PIPES, 25 mM HEPES, 5 mM EGTA, 8 mM MgSO$_4$). The exception was experiments where HEC1 levels were quantified, in which cells were treated with 1x PHEM+0.5% TritonX100 for 5 min prior to PFA. Coverslips were washed with 1x PHEM+0.5%

**Table 2.** Yeast strains used in this study.

**All strains are derivatives of SBY3 (W303)**

| Strain | Relevant genotype |
| --- | --- |
| SBY3 (W303) | *MATa ura3-1 leu2-3,112 his3-11 trp1-1 can1-100 ade2-1 bar1-1* |
| SBY13772 | *MATa STU2-3HA-IAA7:KanMX DSN1-6His-3Flag:URA3 his3::pGPD1-TIR1:HIS3* |
| SBY13901 | *MATa STU2-3HA-IAA7:KanMX DSN1-6His-3Flag:URA3 his3::pGPD1-TIR1:HIS3 leu2::pSTU2-STU2-3V5:LEU2* |
| SBY17069 | *MATa STU2-3HA-IAA7:KanMX DSN1-6His-3Flag:URA3 his3::pGPD1-TIR1:HIS3 leu2::pSTU2-stu2(Δ592–607::GDGAGLlinker)−3 V5:LEU2* |
| SBY17206 | *MATa STU2-3HA-IAA7:KanMX DSN1-6His-3Flag:URA3 his3::pGPD1-TIR1:HIS3 leu2::pSTU2-stu2(K598A R599A)−3 V5:LEU2* |
| SBY17477 | *MATa STU2-3HA-IAA7:KanMX DSN1-6His-3Flag:URA3 his3::pGPD1-TIR1:HIS3 leu2::pSTU2-stu2(K598A)−3 V5:LEU2* |
| SBY17479 | *MATa STU2-3HA-IAA7:KanMX DSN1-6His-3Flag:URA3 his3::pGPD1-TIR1:HIS3 leu2::pSTU2-stu2(R599A)−3 V5:LEU2* |
| SBY17519 | *MATa STU2-3HA-IAA7:KanMX DSN1-6His-3Flag:URA3 his3::pGPD1-TIR1:HIS3 leu2::pSTU2-stu2(Δ560–657::GDGAGLlinker)−3 V5:LEU2* |
| SBY17593 | *MATa STU2-3HA-IAA7:KanMX DSN1-6His-3Flag:URA3 his3::pGPD1-TIR1:HIS3 leu2::pSTU2-stu2(Δ560–657::GDGAGLlinker:592–607:GDGAGLlinker)−3 V5:LEU2* |
| SBY18799 | *MATa STU2-3HA-IAA7:KanMX DSN1-6His-3Flag:URA3 his3::pGPD1-TIR1:HIS3 leu2::pSTU2-Stu2(Δ560–657::chTOG(1081–1167))−3 V5:LEU2* |
| SBY18797 | *MATa STU2-3HA-IAA7:KanMX DSN1-6His-3Flag:URA3 his3::pGPD1-TIR1:HIS3 leu2::pSTU2-Stu2(Δ560–657::linker-chTOG(1137–1150)-linker)−3 V5:LEU2* |
| SBY19023 | *MATa STU2-3HA-IAA7:KanMX DSN1-6His-3Flag:URA3 his3::pGPD1-TIR1:HIS3 leu2::pSTU2-stu2(Δ560–657::chTOG(500-585))−3 V5:LEU2* |

TritonX100 for 5 min, then washed three more times with 1x PHEM + 0.1% TritonX100 over 10 min.

**Table 3.** Human cell lines used in this study.

| Cell line | Parental | Modification 1 | Modification 2 | Source |
| --- | --- | --- | --- | --- |
| SBM004 | HCT116 | CKAP5e44-FKBP-EGFP/CKAP5e44-FKBP-EGFP | | *Cherry et al., 2019* PMID:31058365 |
| SBM033 | 293T | | | *Ding et al., 2013* PMID:23154965 |
| SBM001 | HeLa FlpIn Trex | SV40: LacZ-ZeocinR | | *Etemad et al., 2015* PMID:26621779 |
| SBM044 | HeLa FlpIn Trex | TRE: chTOG$^{WT}$-EGFP SV40: PuromycinR | | This study |
| SBM046 | HeLa FlpIn Trex | TRE: chTOG$^{KK/AA}$-EGFP SV40: PuromycinR | | This study |
| SBM045 | HeLa FlpIn Trex | TRE: chTOG$^{WT}$-EGFP SV40: PuromycinR | hPGK1: EB1-mCherry IRES hygromycinR | This study |
| SBM047 | HeLa FlpIn Trex | TRE: chTOG$^{KK/AA}$-EGFP SV40: PuromycinR | hPGK1: EB1-mCherry IRES hygromycinR | This study |

Cells were blocked for 1–2 hr at room temperature in 20% goat serum in 1x PHEM. Primary antibodies were diluted in 20% goat serum to the following final concentrations/dilution factors: anti-centromere protein antibody or ACA (Antibodies Inc 15–235) 1:600; anti-HEC1 clone 9G3 (ThermoFisher Scientific, MA1-23308) 2 µg/mL; anti-alpha tubulin clone DM1A (Sigma Millipore, T6199) 2 µg/mL; anti-Mad1 (GeneTex, GTX109519) 1:1000, anti-Hec1pSer55 (Jennifer DeLuca, Colorado State University) (1:1000). Coverslips were incubated overnight at 4˚C in primary antibody, then washed four times with 1x PHEM + 0.1% TritonX100 over 10 min. Goat anti-mouse, rabbit, and human secondary antibodies conjugated to AlexaFluor 488, 568, 647 (Invitrogen) were all diluted at 1:300 in 20% boiled goat serum with the exception of goat anti-mouse AlexaFluor647 used to target mouse anti-alpha tubulin where 1:600 dilution was used. Coverslips were washed four times with 1x PHEM + 0.1% TritonX100 over 10 min, then stained for 1 min with 30 ng/mL 4′,6-diamidino-2-phenylindole (DAPI, Invitrogen, D1306) in 1x PHEM. Coverslips were washed two times with 1x PHEM, then immersed in mounting media (90% glycerol, 20 mM Tris [pH = 8.0], 0.5% w/v N-propyl gallate) on microscope slides and sealed with nail polish.

## Microscopy and image analysis

Fixed cell images were acquired on either a Deltavision Elite or Deltavision Ultra deconvolution high-resolution microscope, both equipped with a 60x/1.42 PlanApo N oil-immersion objective (Olympus). Slides imaged on the Elite were collected with a Photometrics HQ2 CCD 12-bit camera, while those imaged on the Ultra were equipped with a 16-bit sCMOS detector. On both microscopes, cells were imaged in Z-stacks through the entire cell using 0.2 µm steps. All images were deconvolved using standard settings. Softworx Explorer 2.0 was used to quantify kinetochore intensities by identifying the maximal ACA intensity within a Z-stack and collecting pixel intensity with a 16-pixel region of interest for the appropriate wavelength, as well as a 36-pixel region encompassing the first region for background subtraction. This region is large enough to capture both inner and outer kinetochore signals. Background intensity was found by subtracting the intensity of the 16-pixel region from the 36-pixel region. This background intensity was then divided by the difference in area (20 pixels) to give background intensity per pixel. This was then multiplied by 16 and subtracted from the initial intensity of the 16-pixel region. Representative images displayed from these experiments are projections of the maximum pixel intensity across all Z images. Intensity of cold-stable astral microtubules was quantified from maximum projections of Z-stacks. A rectangular region containing all kinetochores on the astral side of the mitotic spindle was drawn and measured, then background tubulin signal was measured in three small regions near but outside the astral region. Background signal was multiplied by the ratio of area between the two rectangles and then directly subtracted. Photoshop was used to crop, make equivalent, linear adjustments to brightness and contrast, and overlay images from different channels.

Live-cell TIRF microscopy was performed on a Nikon widefield fluorescence and TIRF microscope equipped with an 100x/1.49 CFI Apo TIRF oil immersion objective (Nikon) and an Andor iXon X3 EMCCD camera. Cells were imaged briefly with dual lasers to visualize expression of EB1-mCherry and when appropriate, chTOG-EGFP. TIRF images for quantification were collected every 300 ms over a 90 s period using only the 561 nm laser to monitor EB1-mCherry. Tiff files for each timepoint were imported into FIJI (*Schindelin et al., 2012*), background subtracted, and ROF denoised prior to semi-automated track analysis with the TrackMate plugin (*Tinevez et al., 2017*) using the DoG Detector and Simple LAP Tracker. Kymographs were generated from background subtracted movies with Fiji (*Schindelin et al., 2012*) using a 7 µm line profile over the entire duration of the experiment.

## Transfection of siRNA

180,000 cells were grown on acid washed coverslips in each well of a six-well culture dish for 12–24 hr. 4–5 µL of siRNA were mixed with 6 µL of Lipofectamine RNAiMAX (Thermo Fisher 13778075) in 200 µL of DMEM containing no additives and incubated for 20 min. This was then added to cells in 1 mL of DMEM containing no additives. After 5–6 hr, culture media was replaced with DMEM with FBS and antibiotics and for induction of transgenes, doxycycline.

## Statistics

GraphPad Prism version 8.4 was used for statistical analysis. Data normality was assessed for all experiments using the D'Agostino and Pearson test. For those with normal distributions, mean values were reported and t tests were used. For data sets failing the normality test, median values were reported, and non-parametric Mann-Whitney tests were performed for comparisons. Each test specifically identified in figure legends.

## Multiple sequence alignments

Fungal proteins related to *Saccharomyces cerevisiae* Stu2 were identified using a PSI-BLAST (*Altschul et al., 1997*) search on NCBI. Multiple sequence alignments of the entire proteins were generated with ClustalOmega (*Sievers et al., 2011*) default parameters and displayed in JalView 1.8 (*Waterhouse et al., 2009*). Eukaryotic basic linker sequences were manually identified and aligned with ClustalOmega for display in JalView.

## Nucleic acid reagents

All plasmids used in this study are described in *Table 1*. Construction of a *LEU2* integrating plasmid containing wild-type *pSTU2-STU2-3V5* (pSB2232) was previously described (*Miller et al., 2016*). *STU2* variants were constructed by mutagenizing pSB2232 as described previously (*Liu and Naismith, 2008*; *Tseng et al., 2008*), resulting in pSB2260 (*Miller et al., 2019*; *pSTU2-stu2(Δ560–657:: GDGAGL$^{linker}$)−3* V5, i.e. *stu2$^{ΔBL}$*), pSB2634 (*pSTU2-stu2(Δ592–607::GDGAGL$^{linker}$)−3* V5, i.e. *stu2$^{ΔPatch}$*), pSB3076 (*pSTU2-stu2(Δ551–657::chTOG$^{1081-1167}$)−3* V5, i.e. *stu2$^{hBL}$*), pSB3075 (*pSTU2-stu2(Δ560–657::GDGAGL$^{linker}$:chTOG$^{1137-1150}$:GDGAGL$^{linker}$)−3* V5, i.e. *stu2$^{hPatch}$*), pSB3089 (*pSTU2-stu2(Δ551–657::chTOG$^{500-585}$)−3* V5, i.e. *stu2$^{h2-3\ Linker}$*), pSB2781 (*pSTU2-stu2(K598A R599A)−3* V5, i.e. *stu2$^{KR/AA}$*), pSB2818 (*pSTU2-stu2(K598A)−3* V5, i.e. *stu2$^{K598A}$*), pSB2819 (*pSTU2-stu2(R599A)−3* V5, i.e. *stu2$^{R599A}$*), and pSB2820 (*pSTU2-stu2(Δ551–657::GDGAGL$^{linker}$:592–607:GDGAGL$^{linker}$)−3* V5). pSB2820 was further mutagenized following the above protocols resulting in pSB2852 (*pSTU2-stu2(Δ560–657::GDGAGL$^{linker}$:592–607:GDGAGL$^{linker}$)−3* V5, i.e. *stu2$^{ΔBL+Patch}$*).

Codon-optimized chTOG$^{WT}$ and chTOG$^{KK/AA}$ were synthesized and sub-cloned into pCDNA3.1-C-EGFP by Genscript (pSB2822 and pSB2823, respectively). Both chTOG$^{WT}$ and chTOG$^{KK/AA}$ -EGFP fusions were cloned into pCDNA5 FRT/TO/puro (pSB2353) (*Etemad et al., 2015*) through PCR amplification and isothermal assembly to generate pSB2860 and pSB2863, respectively. EGFP was excised from pSB2860 and pSB2863 through restriction digestion and replaced with 6-His,3-FLAG via isothermal assembly to generate pSB2976 and pSB2977. EB1 was PCR amplified from mCherry-EB1-8, a gift from Michael Davidson (Addgene plasmid # 55035), sequence for a codon-optimized mCherry with a flexible linker was synthesized as a gBlock (IDT) and both were inserted into a third-generation lentiviral vector, pLPH2 (pSB2998) via isothermal assembly to generate pSB3217. Primers used in the construction of the above plasmids are listed in *Table 1* and plasmid maps are available upon request.

The siRNA targeting chTOG and Hec1 were ordered from Qiagen. Hec1 was depleted with a custom synthesized siRNA sequence (5'-CCCUGGGUCGUGUCAGGAA-3') that targets the 5' UTR and was previously validated (*DeLuca et al., 2011*). Hs_ch-TOG_6 FlexiTube siRNA (Qiagen SI02653588) targets the coding DNA sequence of endogenous chTOG (5'-AAGGGTCGACTCAATGATTCA-3') but not the codon optimized EGFP fusion.

## Generation of yeast strains

*S. cerevisiae* strains used in this study are described in *Table 2* and are derivatives of SBY3 (W303). Standard media, microbial, and genetic techniques were used (*Sherman et al., 1974*). Stu2-3HA-IAA7 was constructed by PCR-based methods (*Longtine et al., 1998*) and is described previously (*Miller et al., 2016*).

## Yeast growth assay

The desired strains were grown overnight in yeast extract peptone plus 2% glucose (YPD) medium. The following day, cells were diluted to OD$_{600}$ ~1.0 from which a serial 1:5 dilution series was made and spotted on YPD+DMSO or YPD+100 µM indole-3-acetic acid (IAA, Sigma-Aldrich I3750) dissolved in DMSO plates. Plates were incubated at 23°C for 3 days.

## Generation of modified human cell lines

All human cell lines used in this study are described in *Table 3*. 400,000 HeLa FlpIn TREX cells (SBM001) were grown in 60 mm dishes for 16 hr. Media was aspirated and replaced with 2.5 mL of DMEM containing no supplements. 3.2 µg of p0G44 (pSB2380) and 1 µg of pSB2860/pSB2863 were mixed with 8 µL of Lipofectamine2000 (Invitrogen 11668027) in 400 µL of DMEM (no supplements) for 20 min and then added to cells dropwise. Six hours after transfection, media was aspirated and replaced with DMEM containing 10% FBS and antibiotics. 48 hr post-transfection, media was supplemented with 2.5 µg/mL puromycin (Thermo Fisher A11138-03) and cells were negatively selected for 3 days. Upon reaching confluence, expression of EGFP fusion proteins was induced by addition of 1 µg/mL doxycycline (Sigma-Aldrich, D9891) and EGFP-expressing cells were positively selected by FACS using a SONY MA900 to sort into media lacking doxycycline. Doubly selected polyclonal populations of chTOG$^{WT}$-EGFP (SBM044) and chTOG$^{KK/AA}$-EGFP (SBM046) were frozen and stored for future experiments.

EB1-mCherry was stably transduced into SBM044 and SBM046 via lentivirus. Assembly of replication deficient viral particles was performed as previously described (*Toledo et al., 2014*). Briefly, pLCH2-EB1-mCherry (pSB3217), pPAX2 (pSB2636), and pMD2.G (pSB2637) were co-transfected into HEK-293T cells using PEI. Virus containing supernatant media were harvested 48 hr post transfection and passed through 0.45 µm filters and frozen at −80℃. Filtered viral supernatant media were added to dishes containing SBM044 and SBM046 and 48 hr later hygromycin B (Invitrogen 10687010) was added at 400 µg/mL. Cells were selected for 5 days to generate SBM045 and SBM047.

## Acknowledgements

We thank Linda Wordeman, Mike Wagenbach, and Juan-Jesus Vicente for HCT116 cells with chTOG endogenously EGFP tagged and help with EB1-tracking; Geert Kops for HeLa FlpIn TREX cells and pCDNA5 FRT/TO vectors; Jennifer DeLuca for Hec1 antibodies; Michael Davidson for EB1-mCherry DNA; the entire Biggins Lab and Chip Asbury for feedback on this manuscript.

## Additional information

### Funding

| Funder | Grant reference number | Author |
|---|---|---|
| Howard Hughes Medical Institute | | Sue Biggins |
| Damon Runyon Cancer Research Foundation | | Matthew P Miller |
| National Institutes of Health | R01GM064386 | Sue Biggins |
| National Institutes of Health | P41GM103533 | Matthew P Miller |

The funders had no role in study design, data collection and interpretation, or the decision to submit the work for publication.

### Author contributions

Jacob A Herman, Conceptualization, Resources, Data curation, Formal analysis, Validation, Investigation, Visualization, Methodology, Writing - original draft, Writing - review and editing; Matthew P Miller, Conceptualization, Resources, Formal analysis, Funding acquisition, Validation, Investigation, Visualization, Methodology, Writing - review and editing; Sue Biggins, Conceptualization, Resources, Supervision, Funding acquisition, Writing - original draft, Project administration, Writing - review and editing

## Author ORCIDs
Jacob A Herman (ID) https://orcid.org/0000-0002-2069-5810
Matthew P Miller (ID) http://orcid.org/0000-0003-2012-7546
Sue Biggins (ID) https://orcid.org/0000-0002-4499-6319

## Decision letter and Author response
Decision letter https://doi.org/10.7554/eLife.61773.sa1
Author response https://doi.org/10.7554/eLife.61773.sa2

## Additional files

### Supplementary files
• Transparent reporting form

### Data availability
All data generated or analyzed during this study are included in the manuscript and supporting files.

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

## Appendix 1

MT assembly rates have been reported to decrease after chTOG depletion by RNAi in some studies (*Ertych et al., 2014*; *van der Vaart et al., 2011*), while no changes were observed in a different study (*Cassimeris et al., 2009*). In our work, we observed increased MT assembly rates. We believe these differences arise from our use of lentivirus to stably express EB1-mCherry from the human PGK1 promoter, while all three other studies transiently over-expressed the EB protein from a highly active CMV promoter. While cells tolerate EB1 over-expression relatively well, high levels of expression and the length of the linkage between EB1 and EGFP have both been shown to alter MT polymerization rates (*Geisterfer et al., 2020*; *Skube et al., 2010*). Because of these differences, we believe the increase in polymerization rates we observe is a result of changes in the cellular pool of free tubulin that increases because chTOG no longer contributes to MT nucleation or the sequestration of free dimers. In addition, altered tubulin homeostasis has been observed with siRNA depletion of numerous kinesins (*Wordeman et al., 2016*) and could contribute to our results.

