## [Decision Letter]

**Acceptance summary:**

Correcting erroneous chromosome attachments during cell division is crucial for proper chromosome segregation. Elegant genetic and cell biological experiments in this paper convincingly demonstrate that human chTOG contributes to correcting chromosome attachments. The data suggest that this is a conserved function of the TOG family of proteins, thus complementing the well-known – and equally conserved – Aurora-B dependent error correction pathway.

**Decision letter after peer review:**

Thank you for submitting your article "chTOG is a conserved mitotic error correction factor" for consideration by *eLife*. Your article has been reviewed by three peer reviewers, one of whom is a member of our Board of Reviewing Editors, and the evaluation has been overseen by Anna Akhmanova as the Senior Editor. The following individuals involved in review of your submission have agreed to reveal their identity: Alex Kelly (Reviewer #2); Stephen J Royle (Reviewer #3).

The reviewers have discussed the reviews with one another and the Reviewing Editor has drafted this decision to help you prepare a revised submission.

We would like to draw your attention to changes in our revision policy that we have made in response to COVID-19 (https://elifesciences.org/articles/57162) and that apply to papers like yours, where the editors judge that the submitted work as a whole belongs in *eLife* but that some conclusions require a modest amount of additional new data.

Specifically, should your lab not be able to carry out experiments at the moment, we are asking that the manuscript be revised to either limit claims to those supported by data in hand, or to explicitly state that the relevant conclusions require additional supporting data.

*eLife*'s expectation is that authors will eventually carry out the additional experiments and report on how they affect the relevant conclusions either in a preprint on bioRxiv, or if appropriate, as a Research Advance in *eLife*, either of which would be linked to the original paper.

Summary:

This paper by Herman, Miller and Biggins proposes a new function for chTOG at kinetochores as a correction factor which prevents mitotic errors. This is a high quality study that will be of wide interest to cell biologists studying mitosis and cell division in cancer.

Prior work has established a role for Stu2, the budding yeast chTOG ortholog, in correcting improper kinetochore-microtubule attachments. This study expands this concept from budding yeast to human cells, and therefore potentially to other eukaryotes as well. Importantly, the authors identify critical residues in both Stu2 and chTOG. Mutation of these residues in chTOG (KK/AA) yielded an interesting separation of function mutant. While cells depleted of chTOG have major defects in spindle formation, cells expressing chTOG-KK/AA had specific defects in chromosome attachment. The type of defects are consistent with a failure in correcting improper kinetochore-microtubule attachments. The Aurora B kinase has a conserved role in correcting improper kinetochore-microtubule attachments, and the authors provide evidence that Aurora B and chTOG act in distinct pathways, which prior work suggested is also the case in budding yeast. The identification of the two important basic residues is an important step forward in understanding the biophysics that underly Stu2/chTOG's function in error correction.

Overall, this story is an impressive contribution to the field, as it demonstrates that Stu2/chTOG plays a conserved role in error correction and suggests that chTOG works independently of Aurora B.

Essential revisions

1) It is not clear whether the basic residues identified in Stu2 are indeed homologous with those identified in chTOG, nor what role they are playing in error correction. Some further clarification is required.

The authors suggest the basic residues act through an interaction with HEC1/Ndc80 in both budding yeast and human cells, but no evidence is presented that the identified basic residues in chTOG interact with HEC1/NUF2, or that the stu2-KR/AA mutant indeed has a defect in error correction. In their previous paper (Miller et al., 2019), crosslinking/MS identified connections between Stu2 and Ndc80/Nuf2. We suggest that the authors use crosslinking/MS or other assays to demonstrate an interaction between the chTOG region around the critical KK residues and HEC1/Ndc80. These experiments would strengthen and support the model drawn in Figure 5C.

2) Unlike in Stu2, the "patch" in chTOG is just N-terminal to the TOG5 domain. A previous study on the *Drosophila* chTOG homolog (Byrnes and Slep, 2017) identified part of this patch region forming a unique N-terminal extension of the TOG5 domain that is conserved in human chTOG, and mutation of TOG5 had similar defects in mitotic delay and chromosome alignment as observed with the chTOG KK/AA mutations used here. This raises the possibility that the KK/AA mutation is perturbing TOG5 function, which was proposed to interact with the microtubule lattice. To help address this, the authors could mutate the core TOG5 domain and assay whether or not it plays a role in error correction.

(While all reviewers considers this an interesting and important experiment, they also agree that it may not be currently feasible. If you cannot carry out this experiment at the current time, please add an appropriate discussion.)

3) The authors propose that the important KK residues identified in chTOG are required to turn over erroneous KT-MT attachments, and that mutation of these residues causes an increase in KT-MT stability. Although the assays used in Figures 4B and 4C are definitely suggestive of this, a cold-stable kinetochore-microtubule assay or visualization of other markers that report on KT-MT attachment stability (e.g. kinastrin, Ska1/3) would be more definitive.

4) In line with their previous results in yeast, the authors present data that chTOG facilitates error correction independently of Aurora B. However, it is surprising that Hec1 Ser55 phosphorylation does not increase in the presence of the numerous spindle and alignment defects caused by chTOG depletion (Figures 3C, 4A, and 5B). This is obviously very intriguing. Can the authors comment on or provide evidence of whether the kinetochores are under tension in the chTOG siRNA condition in Figure 5B? Also, the authors should report the pSer55 intensities on a per kinetochore basis instead of the mean kinetochore intensity per cell. A small population of tensionless attachments with high amounts of pSer55 could be averaged out by the quantification method used in Figure 5B.

5) Results paragraph two and Figure 1E/F: Instead of comparing one condition with MTs to another without, this experiment should be repeated using the thymidine block and nocodazole treatment detailed in Figure 1B to rule out any MT contribution to chTOG kinetochore staining. A blot for chTOG upon Hec1 siRNA should be also included in the supplemental materials.

6) In the mutant condition, the number of MAD1 positive KTs is lower (Figure 4B). On face value this means the cell is happy for mitosis to proceed, but it does not. The data on mitotic progression is restricted to a simple count of mitotic index (Figure 2—figure supplement 1D). It would be good to see exactly where the delay is happening using live cell imaging to look at timing of each stage. Presumably the SAC is active? Would the KK/AA cells progress if the SAC is inactivated artificially (e.g. Mad2 knock-down)?

---

## [Author Response]

Essential revisions1) It is not clear whether the basic residues identified in Stu2 are indeed homologous with those identified in chTOG, nor what role they are playing in error correction. Some further clarification is required.The authors suggest the basic residues act through an interaction with HEC1/Ndc80 in both budding yeast and human cells, but no evidence is presented that the identified basic residues in chTOG interact with HEC1/NUF2, or that the stu2-KR/AA mutant indeed has a defect in error correction. In their previous paper (Miller et al., 2019), crosslinking/MS identified connections between Stu2 and Ndc80/Nuf2. We suggest that the authors use crosslinking/MS or other assays to demonstrate an interaction between the chTOG region around the critical KK residues and HEC1/Ndc80. These experiments would strengthen and support the model drawn in Figure 5C.

We agree that crosslinking mass-spec with the human proteins would strengthen the model; however, purification of these proteins at sufficient quantities is a non-trivial matter. Full length human Ndc80 and chTOG have only been purified in quantities sufficient for single molecule studies and much greater amounts are required for crosslinking mass spectrometry studies. Larger amounts of Human Ndc80 complex can be obtained by purifying a truncation or a complex where members are fused, but these complexes lack regions that might contribute to the chTOG:Ndc80c interaction. Therefore, we took a genetic approach to address the functional conservation of behavior between the basic linker regions of yeast (Stu2) and the human (chTOG) protein by generating chimeric Stu2 proteins and testing their viability. We replaced the Stu2 basic linker with the sequence from the chTOG basic linker and found that this chimera partially complemented loss of endogenous Stu2. Similar to the yeast data, the 15aa human “patch” was sufficient for this partial rescue. in vitro studies suggested the basic linker functions through a non-specific net positive charge, so we further tested this by replacing the Stu2 basic linker with the chTOG sequence between TOG2 and TOG3. Like the basic linker between TOG4 and TOG5, this region is predicted to be disordered and has a similar π (10.1 and 9.7 respectively). This region failed to even partially rescue the degradation of endogenous Stu2. Together, these additional data further support the conservation of the 15 residue basic linker activity between yeast and humans.

2) Unlike in Stu2, the "patch" in chTOG is just N-terminal to the TOG5 domain. A previous study on the *Drosophila* chTOG homolog (Byrnes and Slep, 2017) identified part of this patch region forming a unique N-terminal extension of the TOG5 domain that is conserved in human chTOG, and mutation of TOG5 had similar defects in mitotic delay and chromosome alignment as observed with the chTOG KK/AA mutations used here. This raises the possibility that the KK/AA mutation is perturbing TOG5 function, which was proposed to interact with the microtubule lattice. To help address this, the authors could mutate the core TOG5 domain and assay whether or not it plays a role in error correction.(While all reviewers considers this an interesting and important experiment, they also agree that it may not be currently feasible. If you cannot carry out this experiment at the current time, please add an appropriate discussion.)

Mutating the conserved tubulin binding loops within TOG5 (or any TOG domain) compromises chTOG’s ability to regulate microtubule dynamics (Widlund et al., 2010, Byrnes and Slep, 2017; King et al., MBoC, 2020; Fox et al., MBoC, 2014) thus analyzing this mutant for error correction would be confounded by altered microtubule dynamics. We agree it is an interesting question whether any of the TOG domains contribute to error correction and hope to develop assays that will directly address this in the future but currently do not have these capabilities.

We have mentioned this future direction in the Discussion and also addressed the reviewers’ hypothesis that the conserved “patch” is contributing to TOG5 lattice binding behavior. First we do not believe the patch or basic pair in chTOG are a structured member of HR0 or TOG5 because they reside 14 amino acids upstream of HR0 helices 12 of which are not resolved in the crystal structure from Byrnes and Slep (2017), likely because as predicted they are disordered. Second, functionally, mutation of TOG5 conserved tubulin binding loops or HR0 compromises chromosome alignment like mutation of the basic pair, but unlike the basic pair mutant they also alter microtubule dynamics and/or spindle organization (Byrnes and Slep, 2017), while the basic pair completely separates these activities.

3) The authors propose that the important KK residues identified in chTOG are required to turn over erroneous KT-MT attachments, and that mutation of these residues causes an increase in KT-MT stability. Although the assays used in Figures 4B and 4C are definitely suggestive of this, a cold-stable kinetochore-microtubule assay or visualization of other markers that report on KT-MT attachment stability (e.g. kinastrin, Ska1/3) would be more definitive.

We performed the requested experiment and found that erroneous kinetochore-microtubule attachments on the astral side of the mitotic spindle are resistant to cold induced depolymerization in cells where chTOG was depleted or the basic pair mutant expressed. These results have been added to Figure 4 and provide additional evidence that the erroneous attachments are due to an increase in kinetochore-microtubule stability.

4) In line with their previous results in yeast, the authors present data that chTOG facilitates error correction independently of Aurora B. However, it is surprising that Hec1 Ser55 phosphorylation does not increase in the presence of the numerous spindle and alignment defects caused by chTOG depletion (Figures 3C, 4A, and 5B). This is obviously very intriguing. Can the authors comment on or provide evidence of whether the kinetochores are under tension in the chTOG siRNA condition in Figure 5B? Also, the authors should report the pSer55 intensities on a per kinetochore basis instead of the mean kinetochore intensity per cell. A small population of tensionless attachments with high amounts of pSer55 could be averaged out by the quantification method used in Figure 5B.

We thank the reviewers for this excellent suggestion. We re-analyzed the data in Figure 5B (now Figure 6C) to quantify the signal of pSer55 phosphorylation at every individual kinetochore and found a 20% increase in Hec1 phosphorylation in chTOG depleted cells. As the reviewers suggested, we likely missed this originally by looking at per cell averages because only ~10% of kinetochores in each cell show an elevated phosphorylation status. These data suggest that when chTOG-based error correction is defective, erroneous, low-tension attachments form that are sufficient to activate the Aurora B Kinase, but its activity is not sufficient to destabilize these errors and they persist. We think this explains the phenotypes we observe. The persistence of low-tension attachments generates a spindle assembly checkpoint giving rise to the increased mitotic index, yet this checkpoint signal is weak because few fully unattached kinetochores are present. Thus, mitosis is delayed but eventually cells exit mitosis with severe and lethal chromosome segregation defects. We have included this new analysis of these data in the revised manuscript (Figure 6C) and accordingly adjusted our interpretation of the data and thank the reviewers for these helpful suggestions.

5) Results paragraph two and Figure 1E/F: Instead of comparing one condition with MTs to another without, this experiment should be repeated using the thymidine block and nocodazole treatment detailed in Figure 1B to rule out any MT contribution to chTOG kinetochore staining. A blot for chTOG upon Hec1 siRNA should be also included in the supplemental materials.

We performed the requested experiment and agree it is easier to interpret when the microtubules are depolymerized. The new results are reported in the revised Figure 1 and confirm that there is a population of chTOG on kinetochores that is independent of microtubules.

6) In the mutant condition, the number of MAD1 positive KTs is lower (Figure 4B). On face value this means the cell is happy for mitosis to proceed, but it does not. The data on mitotic progression is restricted to a simple count of mitotic index (Figure 2—figure supplement 1D). It would be good to see exactly where the delay is happening using live cell imaging to look at timing of each stage. Presumably the SAC is active? Would the KK/AA cells progress if the SAC is inactivated artificially (e.g. Mad2 knock-down)?

While live-cell imaging would better inform the nature of the mitotic delay, COVID-19 restrictions severely limited our access to high-resolution microscopes with environmental controls. We therefore used fixed cell analyses to address reviewers’ desire for a more complete understanding of the mitotic phenotype/delay after chTOG depletion/mutation. First, we found that the mitotic delay in chTOG depleted and KK/AA expressing cells was dependent on Mps1 activity, indicating that the SAC is active in these cells. However, we also found that these cells show a significant increase in chromosome segregation errors. We interpret this to mean that erroneous low-tension attachments are prematurely stabilized and appropriately trigger the SAC to cause a prometaphase-like delay. However, as the reviewers indicated, the low Mad1 signal means this SAC delay is weak and the cells eventually exit mitosis with defective attachments. These results have been added to the revised Figure 3.